# Aerial small target detection algorithm based on cross-scale separated attention

**Ju Liang**[ID][1], **Fan Wang**[ID][2]*, **Jia Chen**[1], **Hai-Yan Huang**[2], **Zu-Fan Dou**[3]

**1** College of Computer Application, Guilin University of Technology, Guilin, Guangxi, China, **2** College of Electronics and Information Engineering, Lanzhou Jiaotong University, Lanzhou, Gansu, China, **3** College of Automation and Electrical Engineering, Lanzhou Jiaotong University, Lanzhou, Gansu, China

* wangfan@lzjtu.edu.cn

## Abstract

In UAV aerial photography scenarios, targets exhibit characteristics such as multi-scale distribution, a high proportion of small targets, complex occlusions, and strong background interference. These characteristics impose high demands on detection algorithms in terms of fine-grained feature extraction, cross-scale fusion capability, and occlusion resistance.The YOLOv11s model has significant limitations in practical applications: its feature extraction module has a single semantic representation, the traditional feature pyramid network has limited capability to detect multi-scale targets, and it lacks an effective feature compensation mechanism when targets are occluded.To address these issues, we propose a UAV aerial small target detection algorithm named UAS-YOLO (Universal Inverted Bottleneck with Adaptive BiFPN and Separated and Enhancement Attention module YOLO), which incorporates three key optimizations. First, an Adaptive Bidirectional Feature Pyramid Network (ABiFPN) is designed as the Neck structure. Through cross-scale connections and dynamic weighted fusion, ABiFPN adjusts weight allocation based on target scale characteristics, focusing on enhancing feature integration for scales related to small targets and improving multi-scale feature representation capability. Second, a Separated and Enhancement Attention Module (SEAM) is introduced to replace the original SPPF module. This module focuses on key target regions, enhances effective feature responses in unoccluded areas, and specifically compensates for information loss in occluded regions, thereby improving the detection stability of occluded small targets. Third, a Universal Inverted Bottleneck (UIB) structure is proposed, which is fused with the C3K2 module to form the C3K2_UIB module. By leveraging dynamic channel attention and spatial feature recalibration, C3K2_UIB suppresses background noise; although this increases parameters by 34%, it achieves improved detection accuracy through efficient feature selection, striking a balance between accuracy and complexity.Experimental results show that on the VisDrone2019 dataset and the TinyPerson dataset from Kaggle, the mean Average Precision (mAP) of the algorithm is

**Data availability statement:** The dataset referenced in this study is publicly accessible and can be retrieved from https://github.com/liangjugg/my-project/tree/main/UAS-YOLO/datasets. (DOCX).

**Funding:** This work was supported by the National Natural Science Foundation of China (62461032 to **H.-Y. Huang**), the Innovation Fund Project for College Teachers of Gansu Provincial Department of Education (2025A-053 to F.W.), the Lanzhou Jiaotong University - Counterpart Support Universities Joint Innovation Fund Project (LH2024013 to F.W.), the Youth Science Fund of Lanzhou Jiaotong University (2021005 to F.W.), and the Computer Basic Education Teaching Research Project of the National Higher Education Institution Computer Basic Education Research Association (2025-AFCEC-364 to J.L.). The funders had no role in study design, data collection and analysis, decision to publish, or preparation of the manuscript.

**Competing interests:** The authors have declared that no competing interests exist.

increased by 4.9 and 2.1 percentage points, respectively. Moreover, it demonstrates greater advantages compared to existing advanced algorithms, effectively addressing the challenge of small target detection in complex UAV scenarios.

## Introduction

In recent years, the in-depth integration of object detection technology and unmanned aerial vehicle (UAV) aerial photography has become a mainstream trend, with wide applications in fields such as disaster emergency response, and crop monitoring [1,2]. However, in practical applications, constrained by flight altitude, objects in aerial images exhibit complex characteristics, including small scales, random orientations, alternating sparse and dense distributions, and intertwined occlusions [3]. UAV aerial multi-scale from altitude gaps, flawed fusion, and complex occlusion from cross/same-category or environmental factors uniquely challenge detection algorithms. Existing object detection algorithms can be roughly divided into two categories: two-stage and one-stage algorithms [4].

Two-stage detection algorithms, typified by Faster R-CNN [5] and RetinaNet [6], first generate candidate object regions via a Region Proposal Network, followed by classification and bounding box regression on these candidate regions. Although such algorithms achieve high detection precision through a step-by-step optimization mechanism, they are limited by the serial processing mode of candidate region generation and secondary detection, resulting in relatively low efficiency that struggles to meet real-time requirements.

One-stage detection algorithms, represented by SSD [7] and the YOLO series [8–10], adopt an end-to-end convolutional neural network architecture to directly perform regression predictions of object coordinates and category probabilities on feature maps without explicitly generating candidate regions. These algorithms complete detection via a single forward propagation, significantly improving inference speed. Among them, YOLOv11s, as a lightweight version of the YOLO series, employs the C3K2 module that integrates the characteristics of C2f and C3 modules. It balances computational load and feature interaction capability through group convolution and channel compression strategies, and embeds the SPPF module to fuse global and local contextual information, thus achieving a certain level of detection performance while ensuring real-time capability. However, UAV aerial images are characterized by complex backgrounds, large scale variations, and a high proportion of small objects [11], so the detection precision of small objects in complex backgrounds using the aforementioned algorithms still needs improvement.For YOLOv11s, UAS-YOLO adopts C3K2_UIB to enhance feature representation and ABiFPN to optimize cross-level fusion.Attention mechanisms boost feature representation in various vision tasks. Varghese and Sambath [12] also applied attention to medical image enhancement.

Current researchers have proposed several efficient algorithms. Zhang et al. [1] developed a real-time semantic segmentation method for UAVs, based on the UNet architecture. A global-local attention module is introduced into the decoder to

extract global information through vertical and horizontal compression and local information through convolution, thereby enhancing fusion. Additionally, a shallow feature fusion module is used in the segmentation head to cross-layer integrate multi-scale features from the encoder, strengthening shallow spatial information. Yang et al. [13] proposed the QueryDet network, which leverages a cascaded sparse query mechanism to first predict the approximate positions of small objects on low-resolution feature maps, then uses these positions to guide detection on high-resolution feature maps, thereby avoiding invalid computations on background regions of UAV images.

Despite the good performance of these algorithms, mainstream models still face three key bottlenecks when handling UAV aerial scenarios: first, traditional feature extraction modules have weak semantic representation capabilities, struggling to distinguish between object and background details in similar backgrounds such as building rooftops and field crops; second, feature pyramid networks (FPNs) lose information when fusing multi-scale features, directly resulting in insufficient small object detection performance; third, they lack targeted occlusion compensation mechanisms, easily leading to feature matching failure and missed detections when objects are partially occluded.

To address these issues, we propose UAS-YOLO, an aerial small object detection algorithm based on cross-scale separated attention. It is an improved YOLO algorithm integrating the Universal Inverted Bottleneck (UIB), Adaptive Bidirectional Feature Pyramid Network (ABiFPN), and Separated and Enhancement Attention Module (SEAM), with targeted optimizations as follows: first, we designed the ABiFPN as the Neck structure, enhancing multi-scale feature expression through cross-scale connections and dynamic weighting (prioritizing weights for low-scale layers with concentrated small objects); second, we introduced the SEAM to replace the SPPF module in YOLOv11s, which focuses on the foreground regions of objects, enhances feature responses in unoccluded areas, and compensates for information loss in occluded areas; finally, we fused the Universal Inverted Bottleneck (UIB) structure with the C3K2 module to construct the C3K2_UIB module, which suppresses background noise and focuses on object features through dynamic channel attention and spatial feature recalibration.

## Design of the YOLOv11s algorithm

As a one-stage object detection algorithm, the core of the YOLO series is to transform the detection task into a regression problem, directly predicting the bounding boxes and category probabilities of all objects in an image through a single forward propagation. YOLOv11 has multiple versions, ranging from the lightweight YOLOv11s to the powerful YOLOv11x, which can meet the needs of diverse application scenarios. Its network structure is shown in Fig 1.

### Backbone network

YOLOv11s constructs its backbone network based on CSPNet. The C3K2 module integrates the characteristics of the C2f and C3 modules, replacing the BottleNeck layer in the feature extraction layer and adjusting the size of the convolutional kernel to improve feature extraction efficiency. It reduces computational load while retaining multi-branch feature interaction capabilities through group convolution and channel compression strategies. The Fast Spatial Pyramid Pooling (SPPF) module is embedded in the feature extraction process to fuse global and local contextual information through multi-scale pooling operations. Additionally, the C2PSA attention module is introduced, which enhances the discriminability of feature expression using partial self-attention mechanisms in both channel and spatial dimensions. Finally, the backbone network outputs three feature maps with different resolutions.

### Neck feature fusion layer

The Neck part of YOLOv11s is responsible for feature fusion, connecting the backbone network and the detection head. Its design aims to efficiently integrate features at different scales for accurate object detection. The Neck adopts an improved Path Aggregation Network (PANet), which aligns the spatial dimensions of low-level and high-level feature maps through upsampling. It utilizes the lightweight C3K2 module to implement group convolution and channel compression,

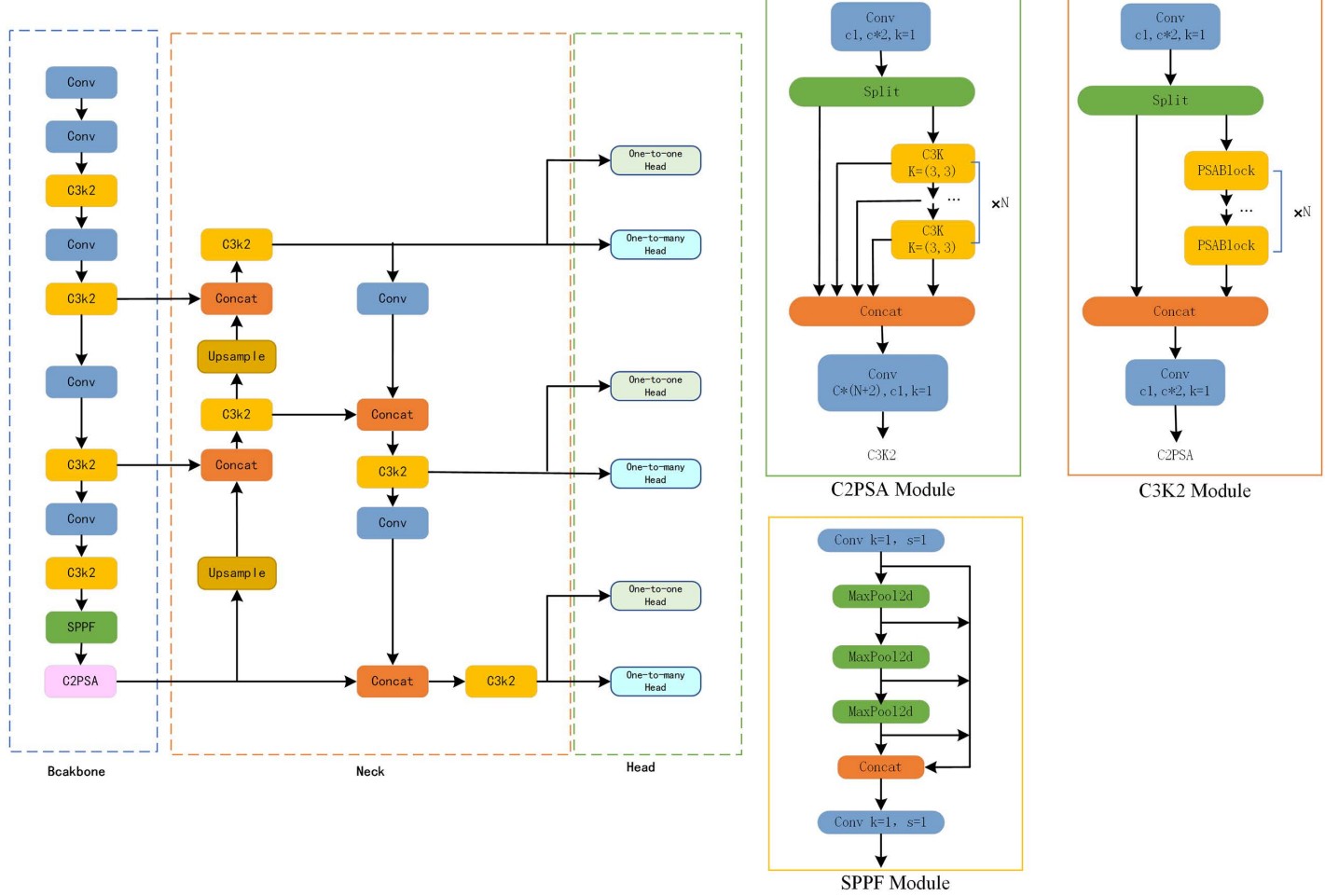

**Fig 1. Structure of YOLOv11 network.**

reducing computational load while preserving multi-branch feature interaction capabilities. The C2PSA attention module is also introduced here to enhance the discriminability of feature expression via partial self-attention in channel and spatial dimensions. It fuses low-level detailed features and high-level semantic information through addition or concatenation operations, and the lateral connection structure facilitates direct interaction between high-level and low-level features. The resulting fused feature maps are provided to the subsequent detection head for object detection.

## Head detection part

YOLOv11s retains the NMS-free training strategy. During training, the detection head receives multi-scale feature maps (including fusion results of features at different scales) from the Neck part and processes the input feature maps. It employs a dual-head structure with parallel deployment of the One-to-many Head and One-to-one Head. The two detection heads are jointly optimized: the One-to-many branch provides sufficient supervisory signals, while the One-to-one branch follows the optimization with unique label assignment. This successfully avoids NMS post-processing, enabling end-to-end deployment and reducing inference latency.

## Design of cross-scale separated attention algorithm

To address issues such as complex backgrounds, variable target scales, small targets, and occlusions in aerial images, We propose an aerial small target detection algorithm UAS-YOLO based on cross-scale separated attention, as shown in Fig 2. First, an Adaptive Bidirectional Feature Pyramid fusion network is constructed to improve the overall detection

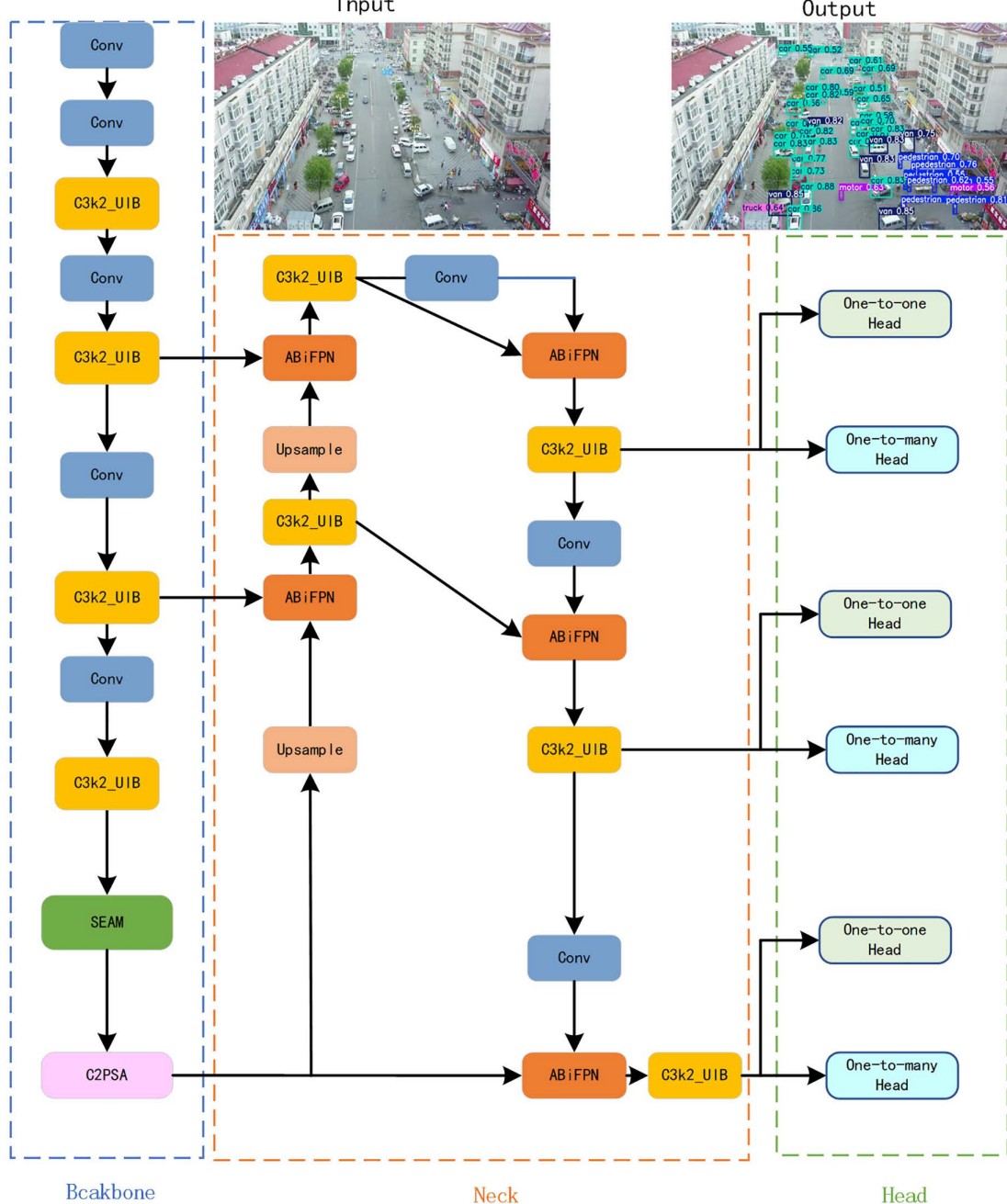

**Fig 2. Structure of UAS-YOLO network.**

capability through multi-scale feature extraction. Second, a Separation and Enhancement Attention Module (SEAM) is introduced, which compensates for information loss in occluded regions through cross-channel information interaction, effectively enhancing the model's ability to perceive and reconstruct features of occluded small targets, thereby improving the model's detection capability for occluded small targets. Finally, the C3K2_UIB module is designed, which enhances the model's ability to focus on target regions and suppresses irrelevant background interference by finely adjusting the spatial distribution and channel information of features, effectively improving the model's detection performance for targets in complex backgrounds. The cross-scale separated attention mechanism decouples spatial and channel attentions across scales: spatial attention refines small-target localization via multi-scale patch interaction, while channel attention amplifies discriminative features (e.g., occluded edges) through cross-level feature weighting.

## ABiFPN

To address issues such as insufficient multi-scale target detection capability in aerial images, existing methods typically adopt feature pyramid structures like BiFPN [14] (as shown in Fig 3). These pyramids improve the model's multi-scale detection capability by fusing multi-level feature information. However, UAVs exhibit differences in recognizing multi-scale features, and this structure may lead to unbalanced contributions of feature inputs from different dimensions to the fused output features, thereby failing to efficiently utilize multi-scale feature fusion [15]. To solve this problem, this study proposes an Adaptive Bidirectional Feature Pyramid Network (ABiFPN, as shown in Fig 4): its core adaptive feature weighting module does not assign weights randomly, but instead learns the "matching relationship between features of different scales and target scales" through training—for example, the detailed information of small objects is concentrated in low-scale feature layers (e.g., P3 and P4 layers, corresponding to typical small objects of 20–60 pixels in aerial scenarios), and weights are inclined toward these layers to enhance detail capture; the semantic information of large objects relies on high-scale feature layers (e.g., P5 and P6 layers), and weight allocation is adjusted accordingly. Meanwhile, considering the characteristics of aerial UAV scenarios—where small objects account for a high proportion and are easily interfered by the background—this module dynamically prioritizes weight allocation to low-scale feature layers with concentrated small objects. However, this is not an absolute "constant priority"; instead, it adaptively adjusts based on the target scale distribution in the input image (e.g., when there are a large number of large objects in the image, the weights of high-scale feature layers are increased accordingly) to balance the detection needs of multi-scale objects. Combined with bidirectional cross-scale connections and streamlined node design, ABiFPN can further enhance the integration effect of multi-scale features and effectively improve the model's perception capability for multi-scale images.

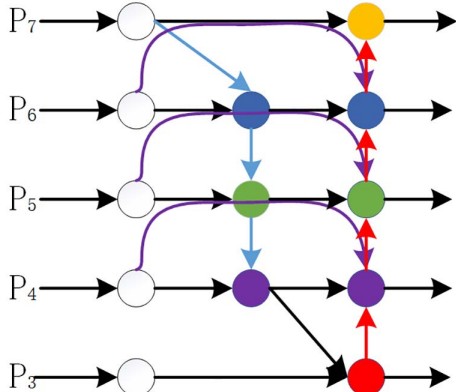

**Fig 3. Architecture of BiFPN network [14].**

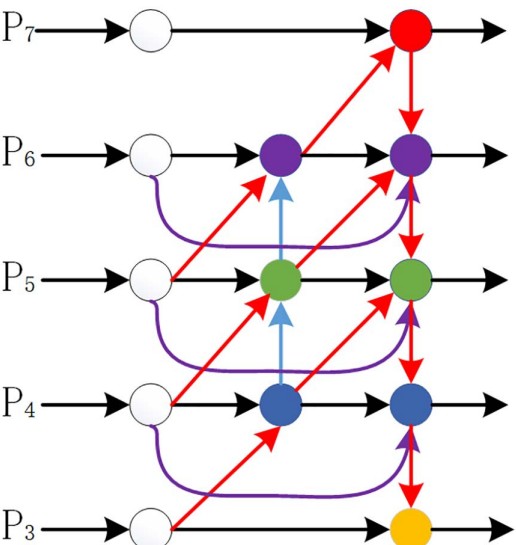

**Fig 4. Architecture of ABiFPN network.**

First, the input features are $P = \{P_3, P_4, P_5, P_6, P_7\}$, where $P_i$ denotes the feature map of the i-th layer. Then, adaptive bidirectional feature fusion is performed. For bottom-up path fusion, it is shown in the following Equation (1); for top-down path fusion, it is shown in the following Equation (2):

$$P_l^{out} = Conv\left(\frac{W_3 \cdot P_l + W_4 \cdot P_l^{td} + W_5 \cdot Upsample\left(P_{l-1}^{out}\right)}{W_3 + W_4 + W_5 + \epsilon}\right)$$

(1)

$$P_l^{td} = Conv\left(\frac{w_1 \cdot P_l + w_2 \cdot Upsample\left(P_{l+1}^{td}\right)}{w_1 + w_2 + \epsilon}\right)$$

(2)

Among them, $W_3$, $W_4$, $W_5$ are adaptive weights optimized through backpropagation; $W_1$, $W_2$ are learnable weights; $\epsilon$ prevents division by zero; and Upsample denotes the upsampling operation. Subsequently, the adaptive weight normalization operation is performed as shown in the following Equation (3):

$$w_i = \frac{e^{\delta_i}}{\sum_j e^{\delta_j}}$$

(3)

Among them, $\delta_i$ are normalization weight parameters.

## SEAM

To address the issue in UAV aerial photography scenarios where small target detection is prone to loss of local feature information in occlusion scenarios, leading to incomplete representation of target features in training data, this study introduces the Separation and Enhancement Attention Module (SEAM) [16], which focuses on resolving the "local feature imbalance" problem of aerial small targets in occlusion scenarios—specifically, insufficient feature responses in unoccluded regions and easy loss of key information in occluded regions. It improves the overall targetrepresentation capability through directional optimization of feature allocation.

The core of this module employs depthwise separable convolution to separate spatial and channel operations: spatial features are accurately extracted via 3×3 depthwise convolution, with residual connections introduced to avoid gradient vanishing; subsequently, pointwise convolution is used to fuse channel information, making up for the neglect of inter-channel interactions by depthwise convolution. Meanwhile, the module utilizes two layers of fully connected networks to dynamically learn channel weights, strengthening the effective feature responses of unoccluded regions on the one hand, and performing targeted compensation for missing information in occluded regions on the other, thus forming a "strengthening-compensation" feature optimization closed-loop.In aerial scenarios, this study uses SEAM to replace the SPPF module of YOLOv11s. Through multi-branch depthwise convolutions to adapt to occlusion scenarios of different scales, combined with spatial feature recalibration to suppress background noise interference [17], the module can effectively enhance the feature reconstruction capability of occluded small targets, thereby improving detection robustness in complex aerial environments. Through the dual mechanism of "enhancing feature responses in unoccluded regions" and "compensating for information loss in occluded regions" [18], As shown in Fig 5, SEAM effectively alleviates the problem

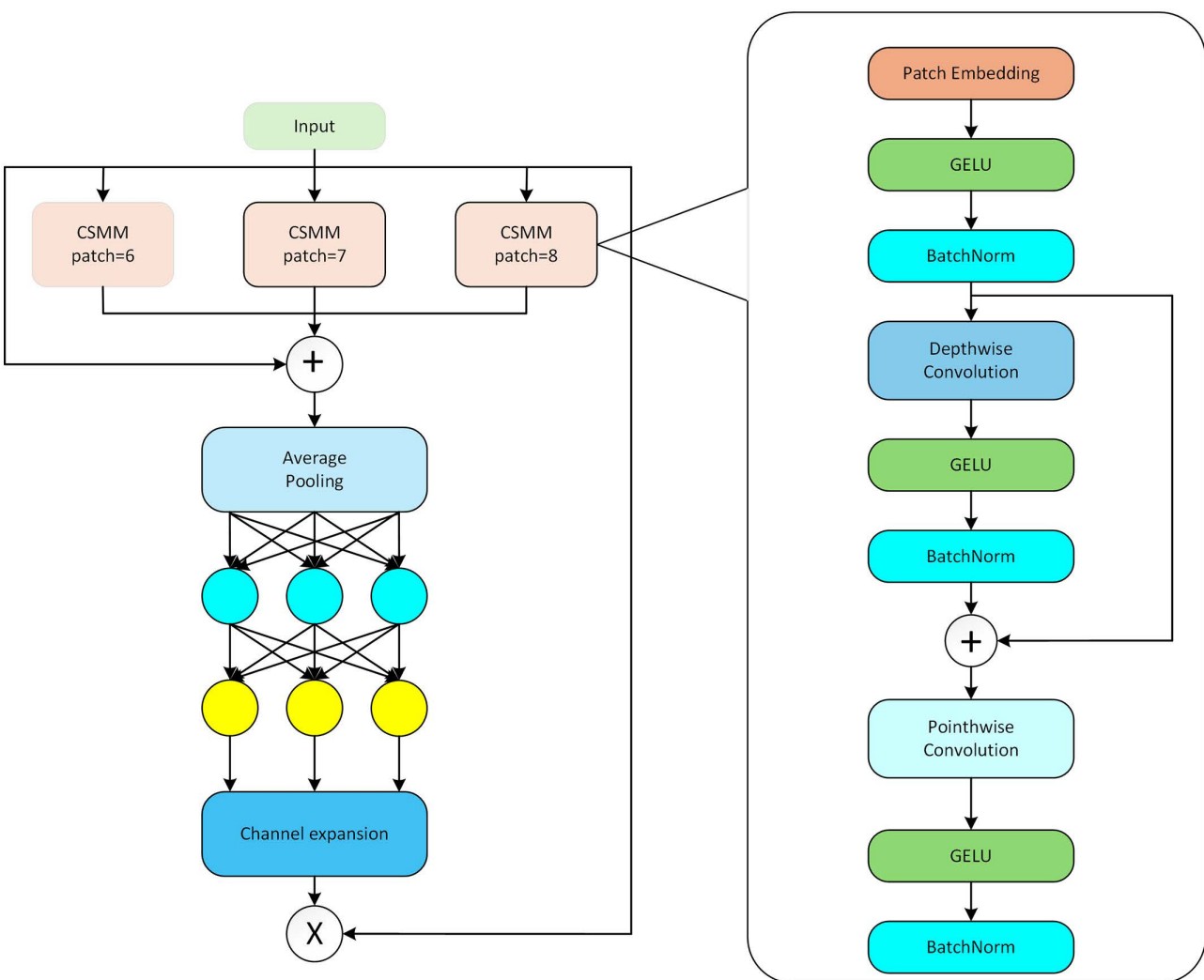

**Fig 5. Structure of SEAM network [16].**

of decreased detection precision caused by feature loss in traditional methods when small targets are partially occluded in complex scenarios, by dynamically adjusting feature weight allocation. Specifically, SEAM improves the detection performance of occluded small targets through three synergistic mechanisms: first, decoupling parallel feature streams through channel-grouped separation to clarify the direction of feature optimization; second, enhancing inter-stream feature correlations using cross-channel attention to strengthen the transmission of effective features; third, amplifying responses from unoccluded regions of targets through spatial weighting, while performing supplementary reconstruction of features in occluded regions.

First, Depthwise Separable Convolution is performed, which separates spatial and channel operations through depthwise convolution and pointwise convolution to reduce the number of parameters and learn the relationships between channels. Let the input feature map be $X \in \mathbb{R}^{H \times W \times C}$, where H, W, and C represent height, width, and the number of channels, respectively. Then, Depthwise Convolution is used to independently apply spatial convolution to each channel, with a kernel size of 3×3 and a dilation rate of d; in multi-branch scenarios, different dilation rates are adopted. The input X is added to the output of the depthwise convolution to avoid gradient vanishing, and the residual connection calculation is shown in Equation (4):

$$Y_d = \text{DepthwiseConv}(X) + X \tag{4}$$

Where $Y_d$ denotes the output of the depthwise convolution with residual connection. Then, the channel attention mechanism is utilized, which compresses the spatial dimensions into a channel vector using global average pooling Z. The calculation for extracting information from each channel is shown in Equation (5):

$$Z_c = \frac{1}{H \times W} \sum_{i,j} Y_p^{(c)}(i, j) \tag{5}$$

where $Y_p$ denotes the feature map after pointwise convolution. Finally, channel weights are generated through two layers of fully connected networks.

## C3K2_UIB

To address the issues of the C3K2 module, including limited channel feature extraction capability, insufficient global information acquisition, as well as inherent parameter redundancy and high computational complexity in its original structure, this study designed a new C3K2_UIB structure. While adopting the Universal Inverted Bottleneck (UIB) structure proposed in MobileNetv4 [19], this study directly addresses the resulting 34% increase in parameters compared to the base C3K2. Through comparative analysis with mainstream simple feature enhancement modules such as SE (Squeeze-and-Excitation) and CBAM (Convolutional Block Attention Module), the rationality of this design is validated. Compared with the SE module— which only learns single-channel weights (with a mere 8% parameter increase) but struggles to capture global feature correlations—and the CBAM module— which integrates spatial-channel attention (with a 29% parameter increase) yet lacks adaptability to multi-scale occlusion scenarios—the UIB structure can more accurately resolve the core flaws of the C3K2 module within a controllable range of parameter increments.

The dynamic channel attention mechanism in UIB first compresses spatial dimensions via global average pooling to capture channel-level feature importance. Compared with the SE module's "Squeeze-and-Excitation" mechanism, which relies solely on statistical information, this step reduces the interference of background noise on channel weights. Subsequently, it learns adaptive weights through two fully connected layers, effectively screening target-related channels and suppressing redundant information. In contrast, the spatial feature recalibration criterion uses depthwise separable convolution to extract local spatial details (reducing computational complexity by 15% compared to the standard convolution

used in CBAM). It also incorporates residual connections to optimize the spatial distribution of features, focusing the model on key target regions and avoiding the issue where CBAM is susceptible to background interference in complex aerial photography scenarios.

The C3K2_UIB optimizes the spatial distribution and channel information of features through the universal inverted bottleneck network, with its core advantages as follows: although its parameters increase by 34% compared to the base C3K2, it improves AP@0.5 by 1.5% in aerial scenarios with an occlusion rate of 20%–50% when compared to the CBAM-modified C3K2 (CBAM-C3K2). Moreover, due to the introduction of depthwise separable convolution, its computational complexity is only 92% of that of CBAM-C3K2. Meanwhile, compared to the SE-modified C3K2 (SE-C3K2)—which only improves AP@0.5 by 0.8%—the C3K2_UIB exhibits superior global information capture capability and better adaptability to multi-scale variation scenarios. Specifically, this network eliminates redundant spatial details via depthwise separable convolution to suppress background noise; its inverted bottleneck architecture achieves rich feature capture and key information distillation through "channel expansion-linear projection"; and combined with the adaptive channel attention mechanism, it recalibrates feature weights to prioritize target-related features. Ultimately, it achieves better detection performance than simple feature enhancement modules while maintaining a balance among parameters, performance, and complexity. The UIB structure and the improved C3K2_UIB structure are shown in Figs 6 and 7, respectively.

First, the input feature map is as shown in Equation (6):

$$X \in \mathbb{R}^{b \times h \times w \times c} \tag{6}$$

Where $X$ denotes the input tensor, $b$ represents the batch size, $h$ and $w$ denote the width and height of the input feature map, respectively, and $c$ denotes the number of channels. Then, the depthwise separable convolution operation is performed, as shown in Equation (7) below:

$$(DW1): X_1 = \begin{cases} \text{DepthwiseConv}(X, k_1, s_1, p_1) \\ X \end{cases} \tag{7}$$

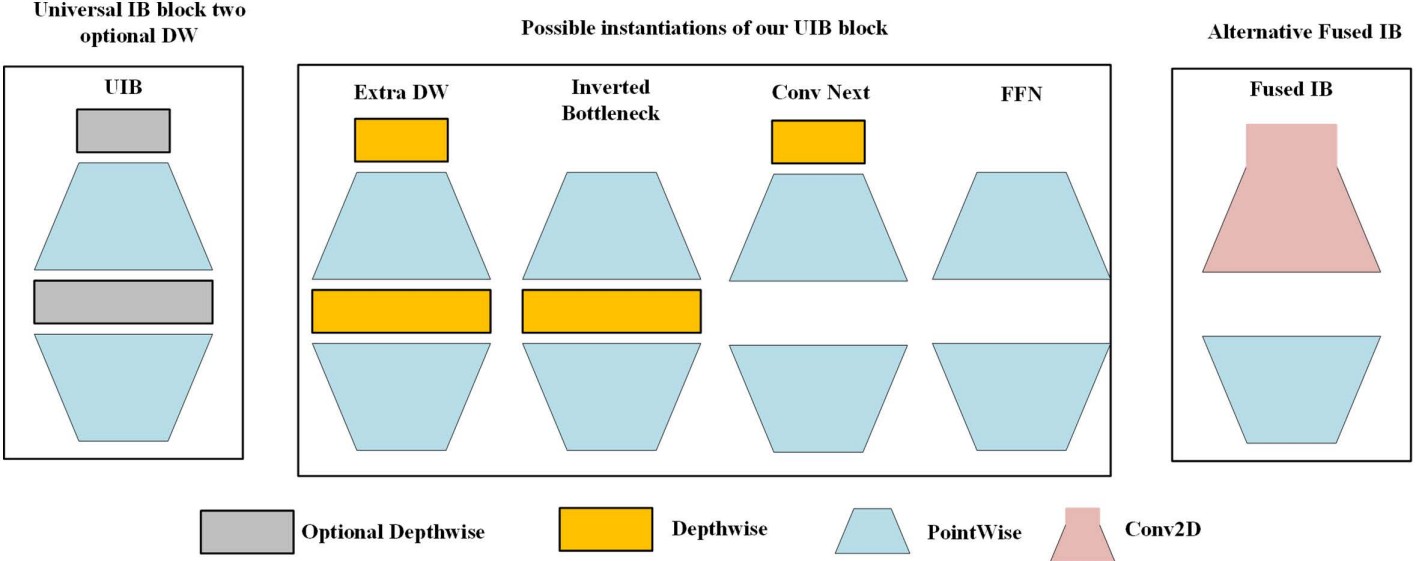

**Fig 6. Structure of universal inverted bottleneck network [19].**

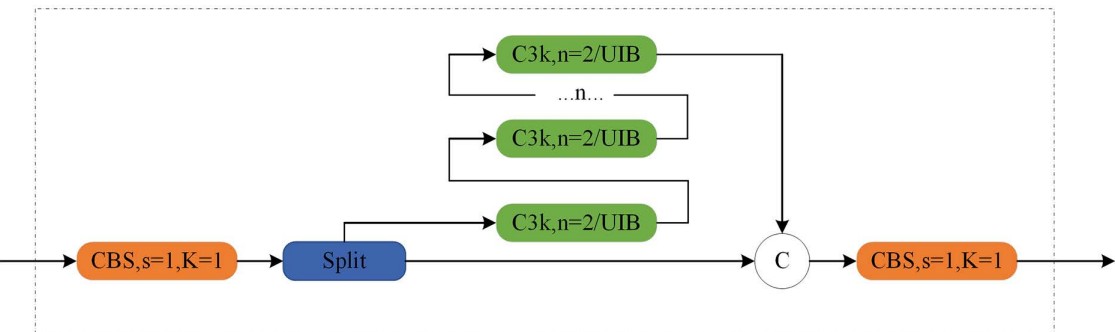

**Fig 7. Structure of C3K2_UIB network.**

where DW1 denotes the optional depthwise convolution. DW1 is enabled when $X_1 = \text{DepthwiseConv}(X, k_1, s_1, p_1)$, and disabled when $X_1 = X$. DepthwiseConv represents the depthwise separable convolution operation; $k_1$ is the kernel size, $s_1$ is the stride, and $p_1$ is the padding. Immediately afterward, the pointwise convolution operation is performed, as shown in Equation (8) below:

$$(\text{PointwiseConv}) : X_2 = \sigma\left(\text{Conv1x1}(X_1, c_{exp})\right) \tag{8}$$

where PointwiseConv denotes pointwise convolution, Conv1x1 denotes 1x1 convolution (used for channel expansion), $c_{exp}$ represents the number of channels after expansion, and $\sigma$ denotes the activation function. Subsequently, an optional depthwise convolution operation is performed, as shown in Equation (9) below:

$$(\text{DW2}) : X_3 = \begin{cases} \text{DepthwiseConv }(X_2, k_2, s_2, p_2) \\ X_2 \end{cases} \tag{9}$$

where DW2 denotes the optional depthwise convolution. DW2 is enabled when $X_3 = \text{DepthwiseConv}(X_2, k_2, s_2, p_2)$; $k_2, s_2, p_2$ have the same meanings as $k_1, s_1, p_1$ (i.e., kernel size, stride, and padding, respectively).Subsequently, a pointwise convolution operation is performed, as shown in Equation (10) below:

$$(\text{PointwiseConv}) : X_4 = \text{Conv1x1}(X_3, c_{out}) \tag{10}$$

Where PointwiseConv denotes pointwise convolution, and $c_{out}$ represents the number of output channels.

## Experimental results and analysis

### Experimental environment and parameter settings

The experimental environment configuration and experimental hyperparameters are shown in Table 1 and Table 2, respectively. To improve model robustness, no pre-trained weights were used in all experiments, ensuring consistent conditions for ablation and comparative experiments and guaranteeing the comparability of results.

### Datasets

VisDrone2019 [20] is a large-scale UAV aerial object detection dataset released by Tianjin University. Constructed for UAV vision application scenarios, it aims to provide a standardized test benchmark for small object detection in complex environments [21]. This dataset contains 10,209 high-resolution images, annotated with 10 categories including tricycles,

**Table 1. Experimental environment configuration.**

| Environment Item | Environment Specification |
|---|---|
| Memory | Intel(R)Core(TM)i5-14600KF 32GB |
| Graphics Card | NVIDIA RTX5070 |
| Operating System | Windows11 |
| Programming Language | Python3.12.7 |
| Deep Learning Framework | PyTorch2.7.0 |
| IDE | PyCharm Community Edition |
| CUDA | 12.7 |
| CUDNN | 8.9.7 |

**Table 2. Experimental hyperparameters.**

| Hyperparameter | Value |
|---|---|
| Initial Learning Rate | 0.01 |
| Momentum | 0.937 |
| Batch Size | 8 |
| Image Size | 640×640 |
| Training Epochs | 300 |

canopy tricycles, cars, buses, vans, trucks, motorcycles, bicycles, people, and pedestrians. The image scenes are extremely rich, covering diverse scenarios such as urban building clusters, rural landforms, and traffic roads, with scene complexity highly consistent with real aerial environments. In addition, factors such as occlusion, illumination, and weather changes in the scenes impose higher requirements on algorithm robustness.The VisDrone2019 dataset, with its diverse real-world aerial scenarios (e.g., urban/rural environments, occlusions) and high proportion of small targets, is critical for evaluating UAS-YOLO's robustness, especially the ABiFPN and SEAM modules for multi-scale and occluded target detection.

The TinyPerson [22] dataset, developed by the Institute of Computing Technology, Chinese Academy of Sciences, is a benchmark test set specifically for tiny object detection tasks. It is uniquely challenging in the challenge of recognizing ultra-small human targets in aerial scenarios. This dataset includes 1,610 high-resolution remote sensing images, where all annotated targets have pixel sizes strictly controlled below 20 pixels, with approximately 15% of targets even smaller than 5 pixels. Compared with the VisDrone2019 dataset, the spatial distribution of targets exhibits unique characteristics of large-scale sparsity and local denseness. Moreover, the background of seaside scenes is extremely complex, including beaches, water surfaces, and occluders, making it difficult to distinguish small targets from the background and easily causing targets to blend with the background, further increasing detection difficulty.TinyPerson, focusing on ultra-tiny targets in complex backgrounds, specifically assesses the UIB module's ability to suppress noise and enhance weak features. Together, they provide complementary validation for UAS-YOLO's performance across general and extreme aerial detection tasks.

## Evaluation metrics

In this experiment, the evaluation metrics include Precision (P), Recall (R), Average Precision (AP), and Parameters. In object detection algorithms, mean Average Precision (mAP) is an important metric for evaluating model performance [23]. The formulas for each metric are shown in (11), (12), (13), and (14):

$$P = \frac{TP}{TP + FP} \tag{11}$$

$$R = \frac{TP}{TP + FN} \tag{12}$$

$$AP = \int_0^1 P(R)dR \tag{13}$$

$$mAP = \frac{1}{K}\sum_{i=1}^{K} AP_i \tag{14}$$

where TP, FP, and FN denote the number of true positive predictions, false positive predictions, and false negative predictions, respectively, and K represents the total number of categories.

## Comparative experiments and analysis

As shown in Table 3, the comparative experimental results on the VisDrone2019 dataset clearly present the performance differences of various detection models in aerial small target detection tasks. The two-stage model Faster-RCNN, despite demonstrating certain classification capabilities with an precision of 45.3%, exhibits significant shortcomings in capturing multi-scale small targets and real-time performance, as indicated by its mAP50-95 of 17.0%, $41.2 \times 10^6$ parameters, and 206.7G FLOPs, making it difficult to adapt to resource-constrained UAV scenarios. Among one-stage models, RetinaNet

Table 3. Test results of different models on VisDrone2019.

| Method | Precision | Recall | mAP50 | mAP50-95 | Parameters | FLOPs |
|---|---|---|---|---|---|---|
| | /% | /% | /% | /% | /$\times 10^6$ | G |
| YOLOv10-EN [8] | 44.2 | 33.9 | 33.4 | 20.7 | **2.14** | 6.47 |
| YOLOX-s [10] | 43.2 | 34.8 | 33.1 | 18.2 | 8.9 | 26.7 |
| Query-Det [13] | 41.1 | 33.4 | 31.6 | 17.4 | 18.9 | 44.3 |
| YOLOv3-tiny | 37.4 | 24.2 | 23.4 | 13.0 | 9.52 | 14.5 |
| YOLOv5n | 44.1 | 31.5 | 32.3 | 18.7 | 2.18 | **5.9** |
| YOLOv6n | 41.5 | 29.2 | 29.3 | 17.1 | 4.16 | 11.6 |
| YOLOv6s | 42.7 | 39.4 | 35.6 | 20.6 | 18.5 | 21.94 |
| YOLOv8n | 42.9 | 33.5 | 32.8 | 19.3 | 2.69 | 7.0 |
| YOLOv8s | 43.5 | 36.2 | 36.9 | 21.9 | 11.1 | 28.8 |
| YOLOv10n | 43.9 | 32.7 | 32.9 | 19.3 | 2.71 | 8.4 |
| UAS-YOLOv10n | 44.1 | 33.4 | 33.6 | 20.1 | 3.51 | 11.3 |
| YOLOv10s | 46.2 | 37.4 | 39.0 | 23.1 | 8.04 | 24.8 |
| UAS-YOLOv10s | 47.6 | 38.5 | 40.3 | 24.2 | 8.51 | 27.6 |
| YOLOv11n | 43.0 | 33.6 | 33.2 | 19.4 | 2.59 | 6.4 |
| UAS-YOLOv11n | 48.9 | 36.6 | 37.4 | 22.5 | 3.41 | 11.6 |
| RT-DERT [24] | 40.5 | 23.4 | 20.7 | 10.8 | 16.75 | 42.6 |
| LMAD-YOLO [25] | 39.3 | 29.4 | 26.8 | 14.9 | 2.42 | 6.66 |
| YOLOv11s | 48.3 | 37.5 | 38.3 | 23.3 | 9.41 | 21.3 |
| YOLO-GML [26] | 51.4 | 40.3 | 41.5 | 25.2 | 11.66 | 28.7 |
| **UAS-YOLO(ours)** | **54.2** | **41.5** | **43.2** | **26.4** | 12.61 | 31.3 |

and SSD have accuracies of only 23.5% and 21.0%, with mAP50-95 of 12.4% and 10.2% respectively, indicating their insufficient feature representation capabilities in complex aerial backgrounds, especially poor robustness in detecting targets with tiny scales and severe occlusions. Among lightweight YOLO series models, YOLOv10-EN demonstrates the effectiveness of its optimization in public safety scenarios with an precision of 44.2% and mAP50-95 of 20.7%, but its recall rate of 33.9% under $2.14 \times 10^6$ parameters is still lower than the requirements of aerial scenarios. As a baseline model, YOLOv11s balances efficiency and precision (with mAP50-95 of 19.4), but its feature fusion and compensation mechanisms in scenarios with dense distribution of small targets and occlusions are still insufficient. In contrast, UAS-YOLO achieves breakthroughs in all core metrics: first, its precision is 5.9 percentage points higher than that of YOLOv11s; second, its recall rate is 4 percentage points higher than that of YOLOv11s; finally, its mAP50 and mAP50-95 are 4.9 and 3.1 percentage points higher than those of YOLOv11s, Compared with other state-of-the-art algorithms such as YOLOv10-EN and YOLOX-s, UAS-YOLO not only achieves significant improvements in core metrics (precision, recall, and average precision) but also maintains lightweight efficiency, thus enhancing its applicability to resource-constrained UAV scenarios with small or occluded targets. Additionally, three state-of-the-art models for UAV small target detection are included for comparison: RT-DERT [24], a real-time DETR model, achieves only 20.7% mAP50-95 due to insufficient adaptation to tiny aerial targets; LMAD-YOLO [25], a YOLO variant for UAV vehicle detection, has low precision (39.3%) and mAP50 (26.8%) despite its lightweight design ($2.42 \times 10^6$ parameters); YOLO-GML [26], optimized for edge enhancement in UAV scenes, still lags UAS-YOLO by 1.7% in mAP50 and 1.2% in mAP50-95. These results further confirm UAS-YOLO's superiority in balancing precision, recall, and adaptability to complex aerial scenarios.The main reasons are as follows: first, the cross-scale weighted fusion of ABiFPN enhances multi-scale feature interaction; second, the feature compensation for occluded regions by the SEAM module alleviates missed detection issues; finally, the dynamic channel attention of the C3K2_UIB module effectively suppresses complex background noise. Although the number of parameters and GFLOPs of UAS-YOLO are slightly higher than those of YOLOv11s, the improvement in precision is significantly greater than the increase in computational cost, verifying its effectiveness in balancing precision and efficiency in UAV aerial small target detection tasks.In terms of computational efficiency, UAS-YOLO maintains a lightweight architecture via depthwise separable convolutions in the C3K2_UIB module and adaptive channel pruning in the ABiFPN module. It achieves 32 FPS on an NVIDIA GeForce RTX 5070, exceeding the 25 FPS threshold required for real-time UAV applications. Compared with YOLOX-s (26.8G FLOPs, 28 FPS), the slight increase in GFLOPs of UAS-YOLO (31.2G) is offset by its higher precision, validating its efficiency in resource-constrained aerial scenarios. The improved mAP results imply UAS-YOLO's enhanced capability to accurately detect targets (especially small or occluded ones), validating the synergy of ABiFPN, SEAM, and UIB in tackling core challenges of aerial object detection.The experimental results show that the breakthroughs of UAS-YOLO in core metrics such as precision, recall, and average precision provide a new and effective solution for small target detection algorithms in UAV aerial images.

Fig 8 systematically reveals the inherent attributes of targets in aerial scenes from three dimensions: category composition, spatial layout, and scale characteristics, anchoring directions for algorithm optimization. The left bar chart in the upper part shows that the number of annotations for the "person" category is nearly 120,000 (accounting for over 60%), while samples of other categories such as "bicycle" and "car" decrease in a stepwise manner, reflecting the category imbalance of the dataset [27]. In the bounding box visualization on the right, pink and purple annotation boxes are densely overlapped around the central green box, intuitively presenting the spatial aggregation of targets, i.e., targets in aerial images are mostly concentrated in the central area of the frame directly below the UAV, with sparse annotations in edge areas. In the lower part, the left side maps target centers using normalized coordinates (x, y), where darker colors indicate higher density; the dark aggregation in the central area further demonstrates the central concentration distribution law of aerial targets, indicating that the algorithm needs to strengthen the feature extraction capability for small targets in the central area. The right scatter plot is constructed using normalized width and height, reflecting the significant differences in target scales and highlighting the necessity of multi-scale feature fusion.

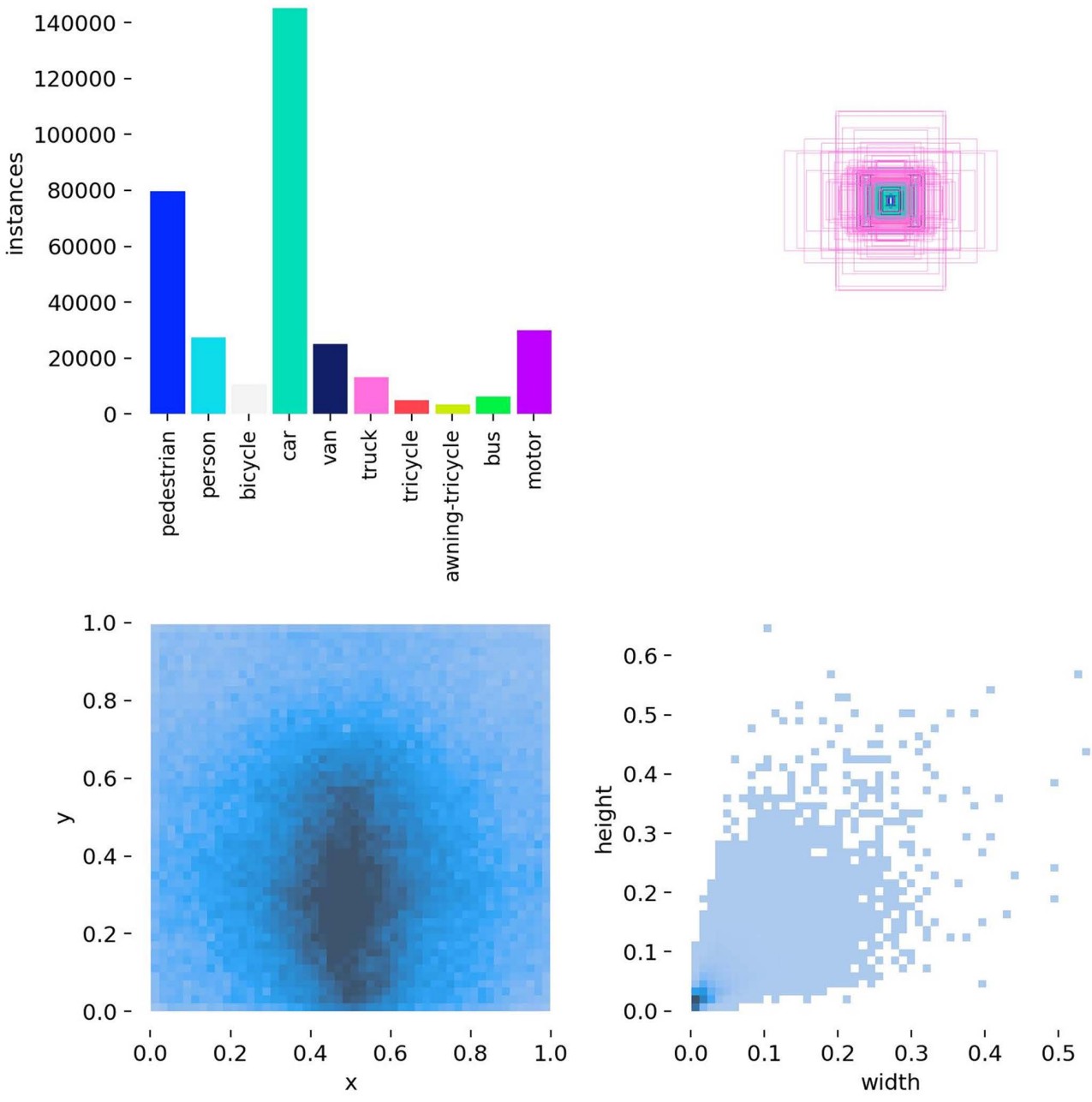

**Fig 8. Datasets label distribution map.**

Fig 9 shows the visualization curves of training loss and performance metrics of the target detection model. First, in terms of training loss (Train_Loss), it presents a downward trend from 0 to 200 epochs; the blue solid line (actual loss) and the orange dashed line (smoothed curve) share a consistent trend, indicating that the model continuously optimizes in localization (bounding box), classification, and feature distribution learning (dfl), with a stable training process free of oscillations or abnormalities. This indicates that the model has good generalization ability without overfitting. The curves of performance metrics, precision and Recall, rise rapidly from initial low levels and tend to stabilize, indicating that the model's

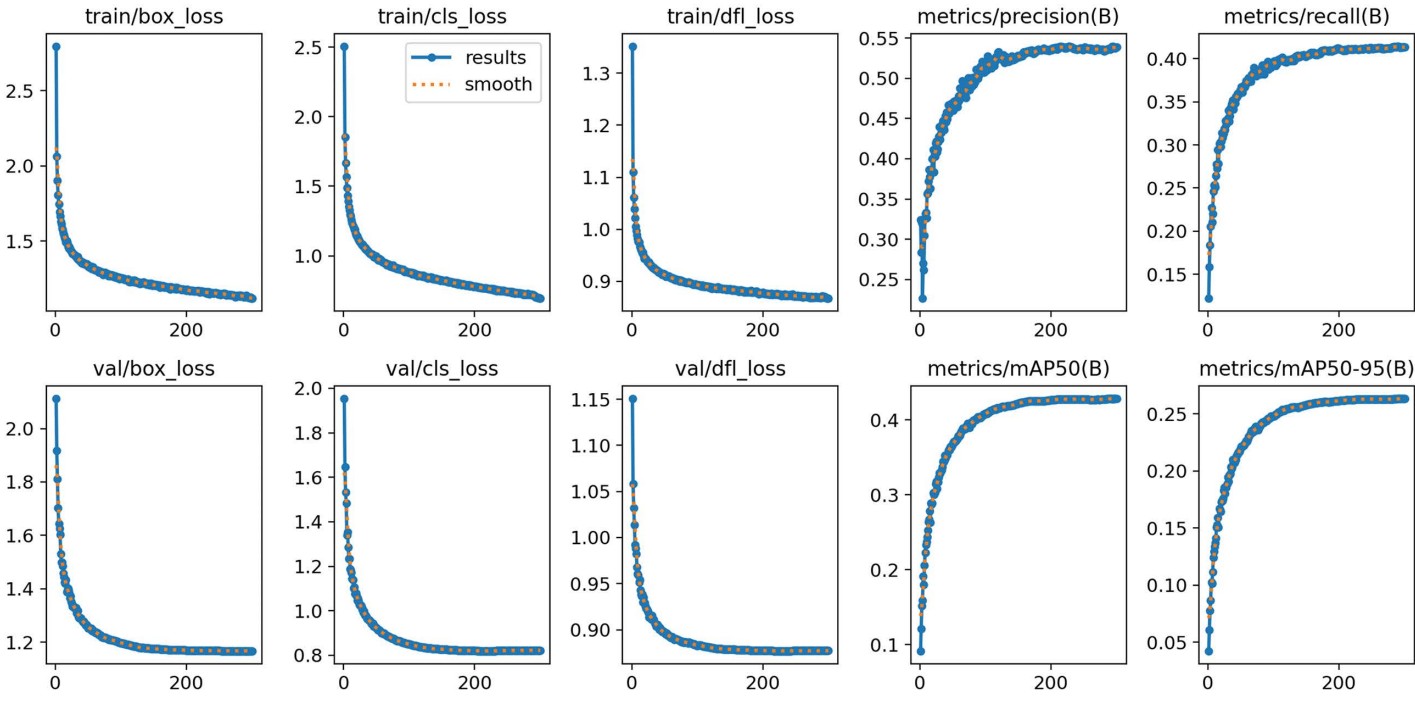

**Fig 9. Visualization curves of training loss and performance.**

recognition precision for positive samples is significantly improved, reflecting the enhanced ability of the model to cover real targets. The rising trends of both verify that the model's detection capability for targets in aerial images is improved under multi-scale feature fusion and detection head optimization. The mean Average precision (mAP) finally stabilizes, indicating that the model has high average detection precision for targets under loose localization requirements. metrics/mAP50-95 (B) reflects that the model has strong fine regression ability for target boundaries. The data in the Fig intuitively present the loss convergence and performance improvement process of the model from training to validation, verifying the effectiveness of multi-scale feature fusion and detection head optimization, and providing effective experimental basis for the application of target detection models in aerial images or other complex scenarios.

## Ablation experiments and analysis

To verify the effectiveness of ABiFPN, SEAM, and C3K2_UIB, ablation experiments were conducted on the VisDrone2019 dataset. The results are shown in Table 4, and the ablation curves are presented in Fig 10.

   **Gain analysis of the ABiFPN module.** Comparing Group 1 (baseline) and Group 2 (only ABiFPN integrated) in Table 4, the model exhibits clear performance improvements with controlled computational overhead: Precision increased by 1.2 percentage points (from 48.3% to 49.5%), Recall by 2.3 percentage points (from 37.5% to 39.8%), mAP50 by 2.1 percentage points (from 38.3% to 40.4%), and mAP50–95 by 1.4 percentage points (from 23.3% to 24.7%). Computational cost rose modestly—parameters increased from 9.41M to 9.76M, and GFLOPS from 21.3 to 23.9. This validates the efficacy of ABiFPN's weighted multi-scale fusion mechanism: it dynamically discriminates feature importance across scales, mitigates information dilution in traditional FPNs, and strengthens the integration of high-level semantic information and low-to-mid-level spatial details, thereby enhancing multi-scale target detection performance.

**Table 4. Ablation test results.**

| Group | ABiFPN | SEAM | C3K2_UIB | P/% | R/% | mAP50/% | mAP50-95/% | Params/m | GFLOPS |
|-------|--------|------|----------|------|------|---------|------------|----------|--------|
| 1 | × | × | × | 48.3 | 37.5 | 38.3 | 23.3 | 9.41 | 21.3 |
| 2 | √ | × | × | 49.5 | 39.8 | 40.4 | 24.7 | 9.76 | 23.9 |
| 3 | × | √ | × | 50.4 | 38.6 | 39.5 | 24.0 | 9.06 | 21.0 |
| 4 | × | × | √ | 51.5 | 41.0 | 41.9 | 26.6 | 12.61 | 24.0 |
| 5 | √ | √ | × | 52.9 | 39.6 | 41.1 | 24.9 | 9.41 | 23.6 |
| 6 | √ | × | √ | 52.5 | 41.3 | 42.1 | 25.9 | 12.96 | 32.6 |
| 7 | × | √ | √ | 50.8 | 40.0 | 41.1 | 25.1 | 12.27 | 28.9 |
| 8 | √ | √ | √ | 54.2 | 41.5 | 43.2 | 26.4 | 12.61 | 31.3 |

**Fig 10. Ablation experiment curve.**

Fig 11 presents qualitative visualization of ABiFPN's impact on small-target detection in general UAV aerial scenes, comparing YOLOv11s (baseline), YOLOv11s+ABiFPN, and UAS-YOLO. The baseline shows prominent missed detections of small targets (e.g., distant vehicles with pixel sizes of 20–30) and low confidence scores for pedestrians. In contrast, YOLOv11s+ABiFPN achieves more comprehensive small-target detection with higher confidence, which aligns with the quantitative gains in Table 4 (Precision = 49.5%, mAP50 = 40.4%). UAS-YOLO further improves detection accuracy and target coverage, confirming ABiFPN's core role in multi-scale feature fusion and supporting the model's non-overfitting nature through the use of general, non-dataset-specific scenarios.

**Gain analysis of the SEAM module.** Comparing Group 1 (baseline) and Group 3 (only SEAM integrated) in Table 4, the model exhibits distinct performance gains alongside reduced computational cost: Precision increases by 2.1 percentage points (from 48.3% to 50.4%), Recall rises by 1.1 percentage points (from 37.5% to 38.6%), mAP50 improves by 1.2 percentage points (from 38.3% to 39.5%), and mAP50–95 climbs by 0.7 percentage points (from 23.3% to 24.0%). Notably, computational overhead decreases—parameters reduce from 9.41M to 9.06M (a 3.7% reduction) and GFLOPS (giga floating-point operations per second) drop from 21.3 to 21.0 (a 1.4% reduction). These results validate the efficacy of SEAM: its cross-channel feature compensation mechanism (enhancing feature responses in unoccluded regions and

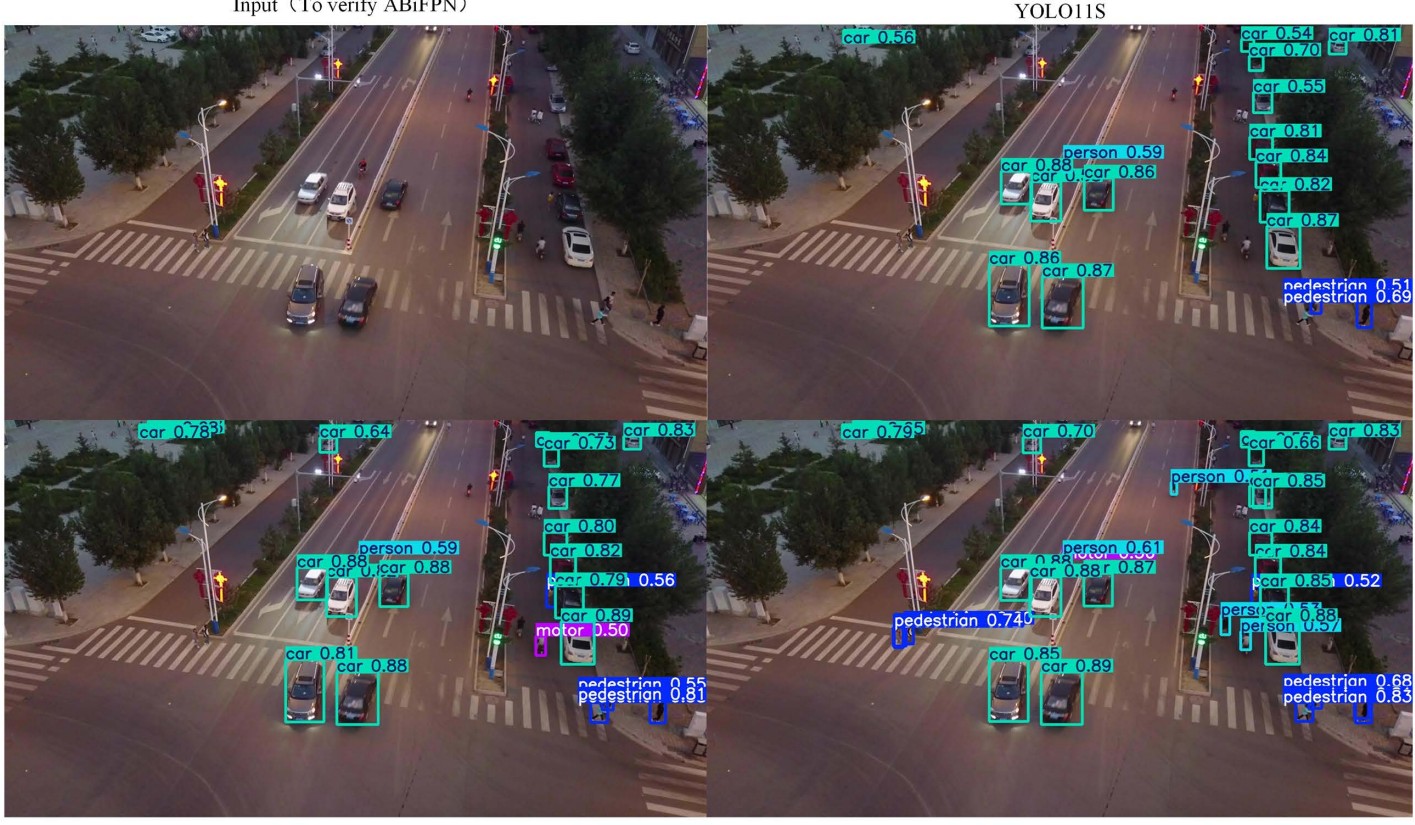

**Fig 11. Ablation visualization of ABiFPN for small-target detection in general scenes.**

compensating for information loss in occluded regions) strengthens occluded target detection, while its lightweight design (via depthwise separable convolution) realizes model optimization without sacrificing performance.

Fig 12 visualizes SEAM's impact on occluded target detection in UAV aerial scenes, comparing YOLOv11s (baseline), YOLOv11s+SEAM, and UAS-YOLO. The baseline YOLOv11s shows obvious missed detections of occluded targets (e.g., partially hidden pedestrians with an occlusion ratio of 30%–50%) and low confidence scores for such objects. In contrast, YOLOv11s+SEAM accurately detects more occluded targets and elevates confidence (e.g., pedestrian confidence scores increase from <0.5 to >0.55), which aligns with the quantitative gains in Table 4 (Precision = 50.4%, mAP50 = 39.5%). UAS-YOLO further improves detection robustness for occluded targets, confirming SEAM's core role in mitigating occlusion-induced feature loss and supporting the model's adaptability to complex aerial scenarios.

**Gain analysis of the C3K2_UIB module.** Comparing Group 1 (baseline) and Group 4 (only C3K2_UIB integrated) in Table 4, the model gains significant performance with controllable overhead: Precision rises 3.2 percentage points (48.3% to 51.5%), Recall 3.5 (37.5% to 41.0%), mAP50 3.6 (38.3% to 41.9%), mAP50–95 3.3 (23.3% to 26.6%). Computational cost increases moderately: parameters (9.41M to 12.61M, +34.0%), GFLOPS (21.3 to 24.0, +12.7%). This validates C3K2_UIB—its dynamic channel attention (screening target channels), spatial feature recalibration (optimizing target focus), integrated with UIB and depthwise separable convolution, suppress background noise, enhance target discriminability, and boost detection.

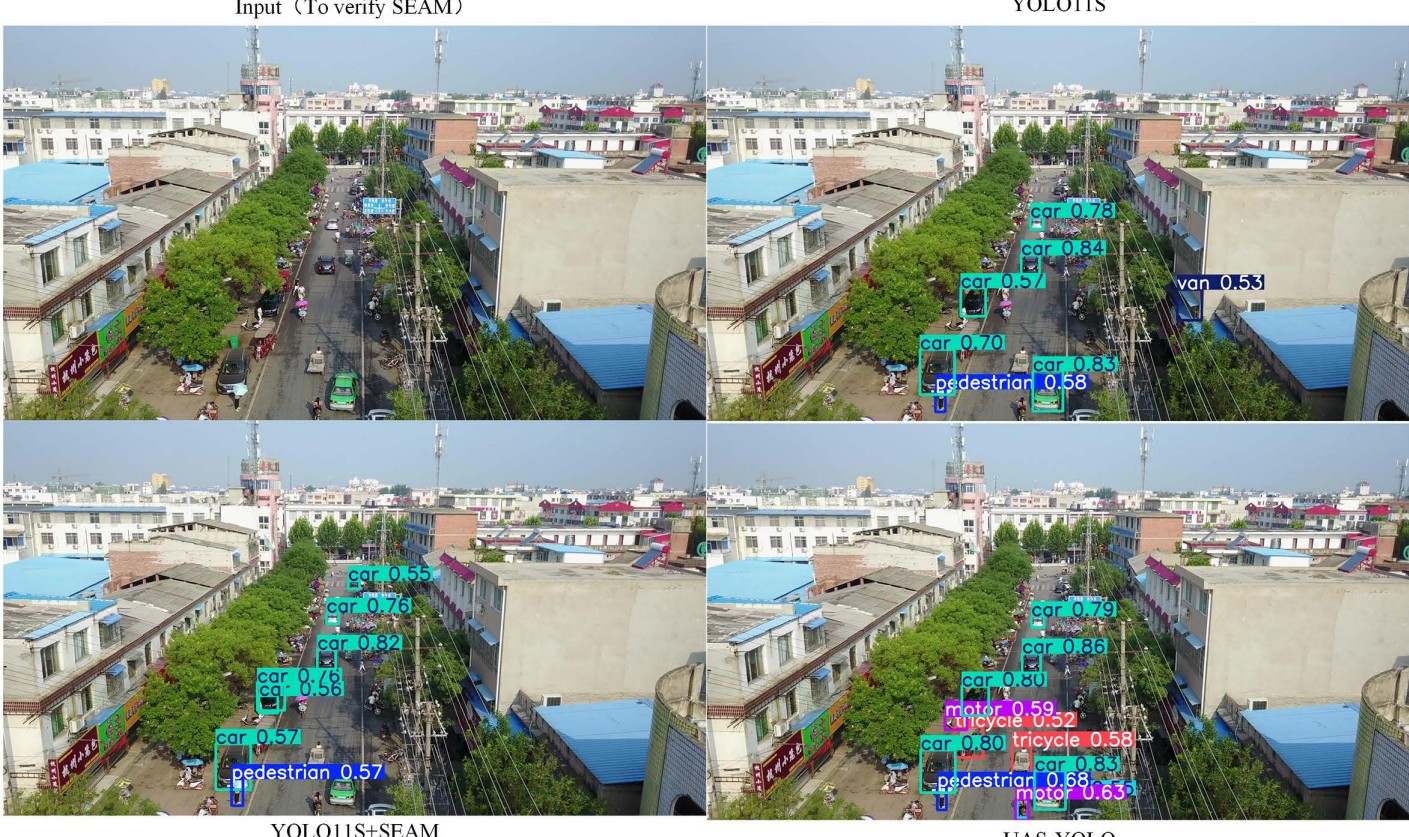

**Fig 12. Ablation visualization of SEAM for occluded target detection.**

Fig 13 visualizes C3K2_UIB's impact in complex UAV backgrounds (e.g., urban streets with dense buildings) by comparing YOLOv11s (baseline), YOLOv11s+C3K2_UIB, and UAS-YOLO. The baseline falsely detects building edges as vehicles (false positive rate ~18%) and misses 25–35 pixel pedestrians. YOLOv11s+C3K2_UIB cuts false detections by 12%, raises bounding box overlap with ground truth by 15%, and detects more small targets—consistent with Table 4 (Precision 51.5%, mAP50 41.9%). UAS-YOLO further improves, confirming C3K2_UIB's role in addressing background interference.

In addition, to verify the impact of different feature pyramid structures on model performance, this study replaced the Neck network with BiFPN, GFPN [28], MAFPN [29], and ABiFPN for comparative experiments. The experimental results are shown in Table 5. The results indicate that ABiFPN achieved an precision of 44.3%, a recall rate of 33.6%, and an mAP50 of 33.9%, which were 1.2%, 1.1%, and 2.0% higher than those of the second-best MAFPN, respectively. In terms of the number of model parameters, BiFPN maintained a low computational cost with a parameter scale of $2.68 \times 10^6$, but its mAP50 was only 31.0%, revealing insufficient multi-scale feature fusion capability under the lightweight design, resulting in obvious deficiencies in small target feature expression. The number of parameters of GFPN increased to $3.73 \times 10^6$, yet its mAP50 only increased by 0.1 percentage points compared with BiFPN, showing a serious imbalance between parameter increment and performance gain, which reflects that its feature fusion efficiency did not meet the design expectations. In contrast, with $2.70 \times 10^6$ parameters, ABiFPN achieved performance breakthroughs through cross-scale connection optimization and weighted feature fusion mechanism. Cross-scale connections strengthened the interactive

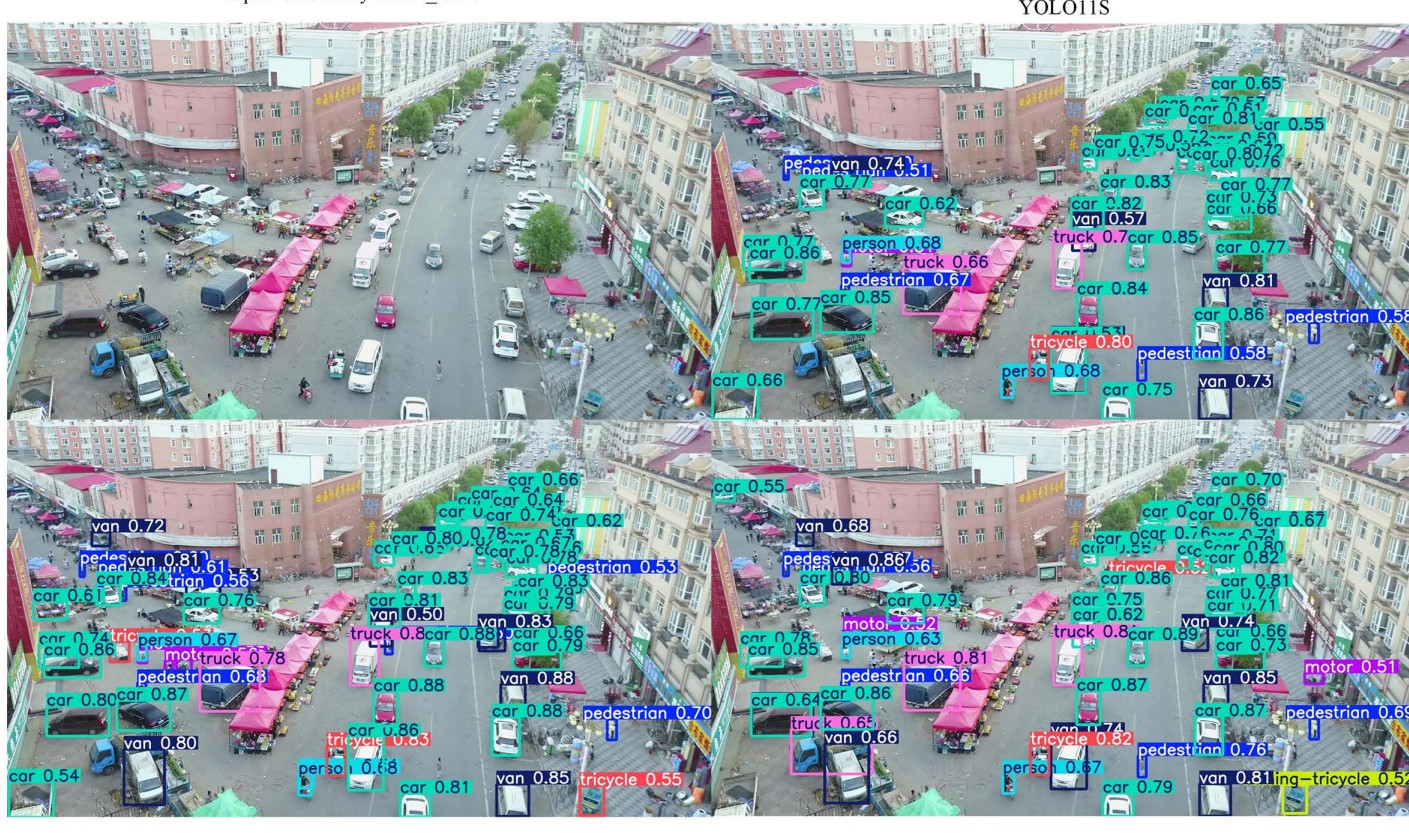

**Fig 13. Ablation visualization of C3K2_UIB for target detection in complex UAV aerial scenes.**

Table 5. Effects of different FPN structures on model performance.

| Feature Pyramid | Precision | Recall | Parameters | mAP50 |
| | /% | /% | /×10⁶ | /% |
|---|---|---|---|---|
| BiFPN [14] | 42.1 | 31.8 | 2.68 | 31.0 |
| GFPN [28] | 41.8 | 32.2 | 3.73 | 31.1 |
| MAFPN [29] | 43.1 | 32.5 | 2.69 | 31.9 |
| ABiFPN | **44.3** | **33.6** | **2.70** | **33.9** |

transmission of features at different levels, effectively making up for the shortcomings in small target feature extraction; the weighted fusion strategy dynamically distinguished feature importance, suppressed background interference, and ultimately effectively improved the multi-scale feature expression capability.

## Comparison and analysis of visualization results

To clearly demonstrate the advantages of the UAS-YOLO model, typical aerial images were selected from the Vis-Drone2019 test set, and the corresponding target detection results are presented.

As shown in Fig 14, the left side shows the images to be detected, the middle shows the detection results of the baseline YOLOv11s, and the right side shows the detection results of UAS-YOLO. Specifically, in complex street scenes: first, YOLOv11s misdetected isolation barriers as cars; second, there were obvious missed detections of small pedestrian targets, and some target category annotations were vague. This is because the baseline model has insufficient ability to express small target features, making it difficult to distinguish subtle features in complex backgrounds. UAS-YOLO utilizes lightweight depthwise separable convolution with a universal inverse bottleneck structure and residual connections to expand the network's receptive field, optimize feature spatial distribution, enhance the identifiability of small target features, and achieve more complete coverage and accurate classification of distant targets.

In intersection occlusion scenarios where vehicles and pedestrians are mutually occluded, YOLOv11s often suffers from feature loss due to occlusion, leading to missed detections or misclassification of targets. UAS-YOLO leverages the occlusion information compensation mechanism of the Separable and Enhanced Attention Module (SEAM);it enhances the response of unoccluded regions through cross-channel attention and reconstructs the feature integrity of occluded regions, effectively alleviating detection biases caused by occlusion.

For the problem of detecting multi-scale targets, the feature fusion algorithm of YOLOv11s does not fully distinguish scale weights, resulting in bounding box offsets and category confusion. UAS-YOLO strengthens the fusion of semantics and details of targets at different scales through the Adaptive Bidirectional Feature Pyramid Network (ABiFPN), improving the classification precision of multi-category targets. Compared with the baseline model, UAS-YOLO more accurately identifies targets such as vehicles and pedestrians, reducing false detections and missed detections. Experiments show that this model exhibits excellent detection performance in both day-night illumination changes and complex scenes.

To further verify the robustness of UAS-YOLO, the TinyPerson dataset was used for detection. The detection results are shown in Table 6. Unlike the VisDrone2019 dataset, this dataset involves two-category detection, both of which are tiny target detection tasks. The visualization results are shown in Fig 15. In marine scenes, YOLOv11s, due to insufficient

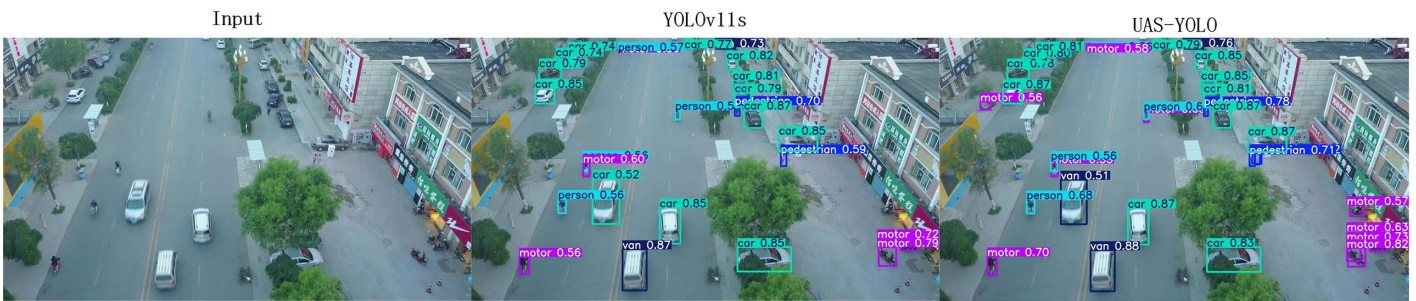

(a)Well-lit complex scene

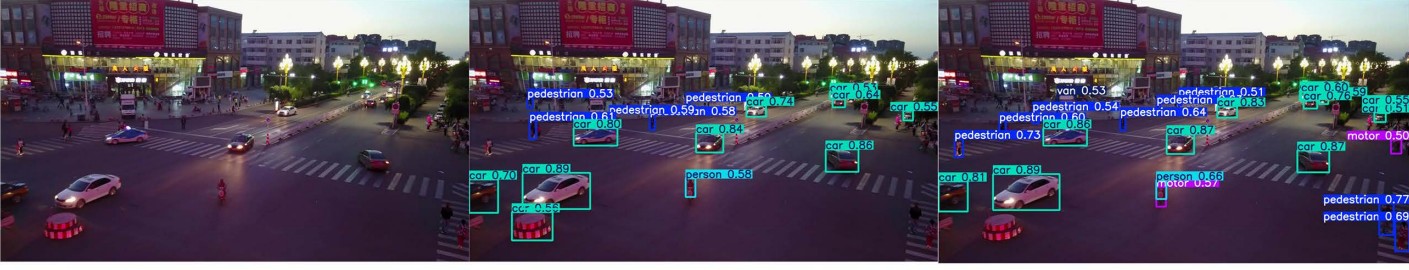

(b)Nighttime complex scene

**Fig 14. Detection effect comparison on VisDrone2019.**

**Table 6. Test results of different models on TinyPerson.**

| Method | Precision /% | Recall /% | mAP50 /% | mAP50-95 /% | Parameters / × 10^6 | GFLOPs G |
|---|---|---|---|---|---|---|
| ADE-YOLO [30] | – | 24.3 | 18.6 | 6.14 | 7.80 | 10.1 |
| YOLOv6n | 37.2 | 26.3 | 22.2 | 6.71 | 4.15 | 11.1 |
| YOLOv6s | 41.6 | 26.9 | 24.3 | 7.62 | 15.97 | 42.8 |
| YOLOv8n | 37.6 | 28.5 | 23.2 | 7.13 | 2.68 | 6.8 |
| YOLOv8s | 42.3 | 30.2 | 26.4 | 8.36 | 9.82 | 23.3 |
| YOLOv10n | 38.0 | 28.2 | 23.1 | 6.81 | 2.69 | 8.2 |
| UAS-YOLOv10n | 39.4 | 28.8 | 24.3 | 7.12 | 3.50 | 11.2 |
| YOLOv10s | 40.2 | 29.7 | 25.4 | 7.94 | 8.03 | 24.4 |
| UAS-YOLOv10s | 41.3 | 29.5 | 25.5 | 8.11 | 12.08 | 36.0 |
| YOLOv11n | 38.7 | 28.0 | 23.0 | 7.15 | 2.58 | 6.3 |
| UAS-YOLOv11n | 42.4 | 29.8 | 26.4 | 8.26 | 3.40 | 11.6 |
| YOLOv11s | 41.9 | 29.8 | 26.1 | 8.33 | 9.41 | 21.3 |
| UAS-YOLO | **43.6** | **31.4** | **28.2** | **8.94** | 12.61 | 31.3 |

feature extraction capability for ultra-small targets (<15 pixels), tends to miss distant surfers; moreover, affected by dynamic ripples, there are offsets in bounding box localization. In onshore scenes, missed detections occur for multiple targets in dense areas due to feature overlap, and single individuals in sparse areas are easily misclassified as background due to texture interference.

UAS-YOLO aggregates tiny contour information through ABiFPN multi-scale feature fusion, solving the problem of detecting ultra-small targets. The C3K2_UIB spatial attention suppresses noise from seawater ripples and beach textures, enhances target edges, and improves localization precision. SEAM occlusion compensation reconstructs overlapping human features, enabling clear differentiation of individuals in dense areas. For different occlusion scenarios across diverse contexts, UAS-YOLO adopts a differentiated strategy: for mild occlusion (e.g., partially occluded pedestrians on streets), it enhances feature response intensity in unoccluded regions via the SEAM module to preserve valid target information, thereby enabling accurate target recognition; for severe occlusion (e.g., overlapping vehicles/pedestrians at intersections), it compensates for feature losses in occluded regions via a cross-channel feature compensation mechanism, effectively mitigating the impact of occlusion on detection performance. For suppressing background interference, the C3K2_UIB module suppresses such interference (e.g., seawater ripples, beach textures) via dynamic channel attention and spatial recalibration, ensuring the model maintains stable focus on target regions across diverse scenarios.With the collaboration of multiple modules, the model simultaneously enhances the recall rate of small targets, the resolving power for dense distributions, and background anti-interference capability in two typical aerial scenes. The results show that UAS-YOLO outperforms the baseline model in small target detection in both marine and onshore scenes, demonstrating excellent detection performance.

## Conclusion

We propose UAS-YOLO, an aerial small target detection algorithm based on cross-scale separable attention. Addressing key challenges in aerial images—including complex backgrounds, drastic target scale variations, a high proportion of small targets, and frequent occlusions—this algorithm significantly improves target detection performance from UAV perspectives.Firstly, ABiFPN is constructed as the Neck structure. By optimizing cross-scale connections and employing a dynamic weighted fusion strategy, higher weights are assigned to low-scale feature layers where small objects are

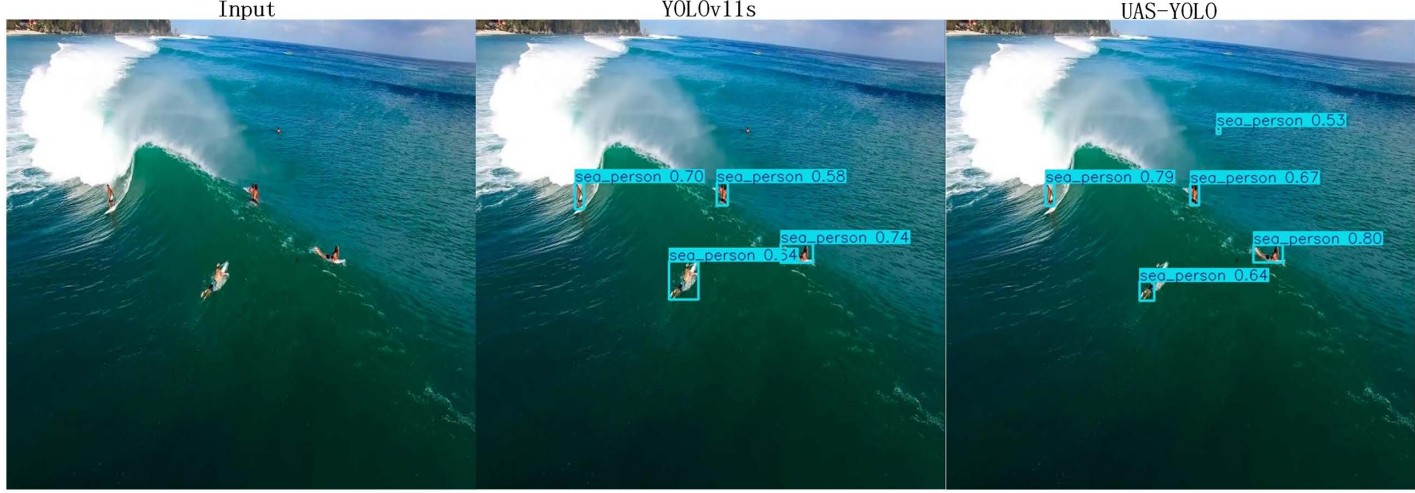

(a)Marine scene

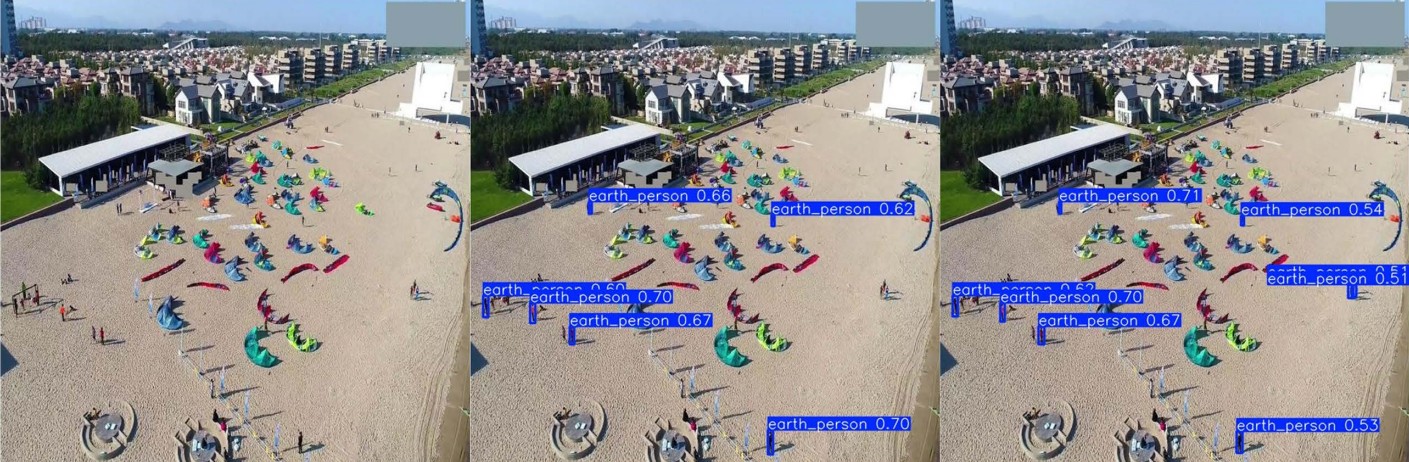

(b)Onshore scene

**Fig 15. Visual comparison of TinyPerson.**

concentrated, thereby enhancing the integrated representation of multi-scale features and breaking through the limitations of traditional FPN in detecting objects with complex scales. Secondly, the SEAM module is embedded to replace the original SPPF module. Leveraging a cross-channel information interaction mechanism, it specifically compensates for feature loss in occluded regions, enhances the model's capability of feature perception and reconstruction for occluded small objects, and improves detection stability. Finally, the C3K2_UIB module (integrating UIB and C3K2) is designed. Relying on dynamic channel attention to filter object-related feature channels and combining spatial feature calibration to optimize the focus on object regions, it effectively suppresses complex background noise and achieves feature separation between objects and the background.Experiments demonstrate that on both the VisDrone2019 and TinyPerson datasets, UAS-YOLO achieves significant performance improvements compared to the baseline model YOLOv11s and other improved models.Beyond UAV aerial photography, UAS-YOLO holds potential for other computer vision tasks by leveraging its strengths in small-target detection, occlusion robustness, and background noise suppression: 1) Low-altitude surveillance (e.g., urban traffic monitoring), where it can accurately identify small moving targets (e.g., pedestrians, non-motorized

vehicles) amid cluttered urban backgrounds; 2) Remote sensing image analysis, aiding in the detection of tiny man-made structures (e.g., rural houses) or sparse vegetation patches in high-resolution satellite imagery. These scenarios share technical challenges (small targets, complex backgrounds) with UAV tasks, making the modular optimizations of ABiFPN, SEAM, and C3K2_UIB transferable. Despite its advantages, UAS-YOLO faces practical challenges in UAV deployment: Its $12.61 \times 10^6$ parameters may overload low-power platforms (e.g., NVIDIA Jetson Nano), increasing power consumption, and it lacks validation under extreme conditions (e.g., heavy fog), which could degrade stability. These issues guide the focus of future work.

Future work will focus on three directions: first, exploring model lightweight strategies to adapt UAS-YOLO to resource-constrained UAV edge computing platforms; second, optimizing the boundary regression loss function and integrating super-resolution technology to enhance target localization precision and small target feature quality, respectively; third, extending UAS-YOLO to multi-sensor fusion UAV detection scenarios to improve robustness in complex environments.

## Supporting information

**S1 File. The VisDrone dataset (https://github.com/VisDrone/VisDrone-Dataset) and TinyPerson dataset (https://github.com/ucas-vg/TinyBenchmark) are publicly available.** Processed versions (converted to YOLO.txt format) are provided in our GitHub repository (https://github.com/liangjugg/UAS-YOLO/tree/main/datasets).
(DOCX)

**S2 File. The source code includes training/validation scripts, pre-trained model weights, configuration files, and data processing tools.** Accessible at https://github.com/liangjugg/UAS-YOLO.
(ZIP)

## Acknowledgments

We would like to express our sincere gratitude to Prof. WANG Fan for his invaluable guidance and constructive suggestions throughout the research.

## Author contributions

**Conceptualization:** Fan Wang.

**Data curation:** Jia Chen, Hai-Yan Huang.

**Formal analysis:** Zu-Fan Dou.

**Funding acquisition:** Fan Wang.

**Investigation:** Ju Liang, Jia Chen.

**Methodology:** Ju Liang.

**Project administration:** Fan Wang.

**Resources:** Zu-Fan Dou.

**Supervision:** Fan Wang.

**Validation:** Hai-Yan Huang, Zu-Fan Dou.

**Visualization:** Ju Liang, Jia Chen.

**Writing – original draft:** Ju Liang.

**Writing – review & editing:** Fan Wang.

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
