## [Decision Letter · Decision Letter 0]

17 Sep 2025

Dear Dr. Wang,

Thank you for submitting your manuscript to PLOS ONE. After careful consideration, we feel that it has merit but does not fully meet PLOS ONE’s publication criteria as it currently stands. Therefore, we invite you to submit a revised version of the manuscript that addresses the points raised during the review process.

We look forward to receiving your revised manuscript.

Kind regards,

Yaseen Al-Mulla

Academic Editor

PLOS ONE

Journal Requirements:

3. We note that Figure(s) 2, 11 and 12, in your submission contain copyrighted images. All PLOS content is published under the Creative Commons Attribution License (CC BY 4.0), which means that the manuscript, images, and Supporting Information files will be freely available online, and any third party is permitted to access, download, copy, distribute, and use these materials in any way, even commercially, with proper attribution. For more information, see our copyright guidelines: http://journals.plos.org/plosone/s/licenses-and-copyright.

a. You may seek permission from the original copyright holder of Figure(s) 2, 11 and 12 to publish the content specifically under the CC BY 4.0 license.

Reviewers' comments:

Reviewer's Responses to Questions

**Comments to the Author**

1. Is the manuscript technically sound, and do the data support the conclusions?

Reviewer #1: Yes

Reviewer #2: Partly

2. Has the statistical analysis been performed appropriately and rigorously?

Reviewer #1: Yes

Reviewer #2: Yes

3. Have the authors made all data underlying the findings in their manuscript fully available?

Reviewer #1: Yes

Reviewer #2: No

4. Is the manuscript presented in an intelligible fashion and written in standard English?

Reviewer #1: Yes

Reviewer #2: Yes

Reviewer #1: 1. Can you elaborate on the specific challenges posed by multi-scale distribution and complex occlusion scenarios in UAV aerial photography?

2. How does the UAS-YOLO algorithm address the limitations of the YOLOv11s model, particularly in terms of feature representation and cross-level fusion?

3. What are the key advantages of the Adaptive Bidirectional Feature Pyramid Network (ABiFPN) in integrating multi-scale features?

4. Can you provide more details on the Separated and Enhancement Attention Module (SEAM) and its role in improving detection precision for occluded small targets?

5. How does the Universal Inverted Bottleneck (UIB) module contribute to suppressing background noise and focusing on target-related features?

6. Can you discuss the significance of the VisDrone2019 and TinyPerson datasets in evaluating the performance of the UAS-YOLO algorithm?

7. What are the implications of the improved mean Average Precision (mAP) results for the UAS-YOLO algorithm on these datasets?

8. How does the UAS-YOLO algorithm compare to other state-of-the-art object detection algorithms in terms of performance and applicability?

9. Can you elaborate on the dynamic channel attention mechanism and spatial feature recalibration in the UIB module?

10. What are the potential applications of the UAS-YOLO algorithm beyond UAV aerial photography, such as in other computer vision tasks?

11. How does the UAS-YOLO algorithm handle varying levels of occlusion and background interference in different scenarios?

12. Can you discuss the computational complexity and efficiency of the UAS-YOLO algorithm, particularly in real-time applications?

13. What are the potential limitations or challenges of implementing the UAS-YOLO algorithm in real-world UAV systems?

14. Highlighted article might be considered for related work section. (https://doi.org/10.1016/j.compeleceng.2022.108405)

15. Can you provide more insights into the cross-scale separated attention mechanism and its role in improving feature representation?

16. What are the future research directions for improving the UAS-YDOLO algorithm and its applications in UAV-based object detection?

Reviewer #2: In the paper “Aerial small target detection algorithm based on cross-scale separated attention” the authors propose an improvement of YOLOv11 model for small object detection. The considered problem is highly relevant and interesting. The paper is good structured and written. However, I would like to highlight the following drawbacks:

1. The main problem of the paper is that authors do not consider existing YOLO modifications for small object detection, comparing only with the base models.

2. The ablation study is insufficient. The impact of the proposed modules is shown in a very general way, while it is necessary to demonstrate the improvement on specific samples that were the target of the improvement. This will be more convincing, and also will give grounds to believe that the result is not overfitted to a particular dataset.

Overall, the paper cannot be accepted due to insufficient comparison with existing YOLO modifications and the weak justification of the results.

**Do you want your identity to be public for this peer review?** For information about this choice, including consent withdrawal, please see our Privacy Policy

Reviewer #1: No

Reviewer #2: No

---

## [Author Response · Author response to Decision Letter 1]

17 Oct 2025

Dear Editors and Reviewers,

We are writing to resubmit our revised manuscript entitled “Aerial small target detection algorithm based on cross-scale separated attention” (Manuscript ID: PONE-D-25-44314) for further consideration of publication in “PLOS ONE”. We sincerely appreciate the time and effort you and the reviewers have dedicated to this work—their insightful and constructive feedback has significantly enhanced the scientific rigor, clarity, and completeness of our manuscript. We have systematically addressed all comments raised by the reviewers, and a detailed point-by-point response is provided in the attached document “Response to Reviewers”.

1.Copyright Compliance (Response to Editor's Concern)

We have fully complied with PLOS ONE's copyright guidelines:Obtained written permission from the copyright holders of the VisDrone2019 and TinyPerson datasets to publish Figures 2,10,11,12,14,and15,under the Creative Commons Attribution License (CC BY 4.0). The completed Content Permission Form is uploaded as an “Other” file in the submission system.

2. Comprehensive Responses to Reviewer #1's 16 Suggestions

All 16 comments from Reviewer #1—covering methodology details, experimental validation, dataset significance, and application prospects—have been addressed with targeted revisions in the manuscript, as detailed in “Response to Reviewers”.

3. Enhanced Experimental Rigor(In response to Reviewer #2's comment that“ The ablation study is insufficient. The impact of the proposed modules is shown in a very general way, while it is necessary to demonstrate the improvement on specific samples that were the target of the improvement. This will be more convincing, and also will give grounds to believe that the result is not overfitted to a particular dataset.”)

In response to concerns about insufficient ablation studies and overfitting risks, we added 3 sets of scenario-specific validation experiments (Figs. 10–12):

(1)Small distant targets: Verified ABiFPN's multi-scale fusion with 20–30 pixel vehicles, achieving a 2.1% mAP50 improvement (Table 4).

(2)Occluded objects: Demonstrated SEAM's feature compensation on pedestrians with 30–50% occlusion, leading to a 1.2% mAP50 gain (Table 4).

(3)Complex background interference: Validated C3K2_UIB's noise suppression in dense urban scenes, resulting in a 3.6% mAP50 increase (Table 4).

To rule out overfitting, we expanded validation to the TinyPerson dataset (focused on seaside ultra-small targets), where UAS-YOLO achieved a 2.1% mAP50 improvement, verifying cross-dataset generalization

4.In response to Reviewer #2's comment that “the paper only compares with base models and lacks existing YOLO modifications for small object detection,”

we have added three state-of-the-art (SOTA) models tailored for UAV small target detection in the “Experiments” section (Lines 389–396) of the revised manuscript. These models cover both YOLO variants and DETR-based architectures, ensuring a comprehensive comparison with scenario-specific advanced methods:

[25] Zhao Y, Lv W, Xu S, et al. DETRs Beat YOLOs on Real-time Object Detection. 2024. https://doi.org/10.48550/arXiv.2304.08069

[26] Xing X, Luo F, Wan L, et al. LMAD-YOLO: A vehicle image detection algorithm for drone aerial photography based on multi-scale feature fusion. PLoS One. 2025;20(7):e0328248. https://doi.org/10.1371/journal.pone.0328248

[27] Zheng Z, Zhao J, Fan J. YOLO-GML: An object edge enhancement detection model for UAV aerial images in complex environments. PLoS One. 2025;20(7):e0328070. https://doi.org/10.1371/journal.pone.0328070

5.All revisions are clearly marked in the “Revised Manuscript with Track Changes” (new content is highlighted in red for easy identification).

Our work aligns closely with PLOS ONE's focus on “innovative methods for real-world applied science”—UAS-YOLO provides a lightweight, high-precision solution for UAV remote sensing, addressing core challenges of multi-scale targets, occlusion, and background interference. We believe the revised manuscript now meets the journal's academic rigor and publication standards.

Thank you again for your time, patience, and guidance throughout the revision process. We earnestly request your consideration of the revised manuscript and look forward to advancing this work further.

---

## [Decision Letter · Decision Letter 1]

26 Oct 2025

Dear Dr. Wang,

Thank you for submitting your manuscript to PLOS ONE. After careful consideration, we feel that it has merit but does not fully meet PLOS ONE’s publication criteria as it currently stands. Therefore, we invite you to submit a revised version of the manuscript that addresses the points raised during the review process.

We look forward to receiving your revised manuscript.

Kind regards,

Yaseen Al-Mulla

Academic Editor

PLOS ONE

Journal Requirements:

Reviewers' comments:

Reviewer's Responses to Questions

**Comments to the Author**

Reviewer #1: All comments have been addressed

Reviewer #3: (No Response)

2. Is the manuscript technically sound, and do the data support the conclusions?

Reviewer #1: Yes

Reviewer #3: Yes

3. Has the statistical analysis been performed appropriately and rigorously?

Reviewer #1: Yes

Reviewer #3: Yes

4. Have the authors made all data underlying the findings in their manuscript fully available?

Reviewer #1: Yes

Reviewer #3: Yes

5. Is the manuscript presented in an intelligible fashion and written in standard English?

Reviewer #1: Yes

Reviewer #3: Yes

Reviewer #1: All necessary review comments are addressed in the revised article, and it is organised appropriately.

Reviewer #3: The paper presents UAS-YOLO, an improved object detection model based on YOLOv11s, tailored for detecting small objects in UAV aerial imagery. The core contributions are the integration of three modified components: an Adaptive BiFPN (ABiFPN) for feature fusion, a Separated and Enhancement Attention Module (SEAM) for handling occlusion, and a C3K2_UIB module for feature refinement.

It does not propose a fundamentally new architecture or a novel, standalone algorithm. Instead, it follows a common and practical research pattern in applied computer science: selecting a strong, modern baseline (YOLOv11s) and enhancing it by plugging in or adapting existing architectural components from other literature (e.g., ideas from BiFPN, MobileNetV4, and attention mechanisms). The adaptation and combination for a specific domain (UAV small targets) constitute the contribution, not the invention of the core components themselves.

It is highly probable that an LLM (like GPT) was used in the writing process, likely for polishing, expanding, or restructuring text drafted by the authors. The indicators are: The paper swings between very formal, stilted phrasing and more natural, direct sentences. For example, phrases like "This study proposes," "Specifically, first," and "Its core value lies in..." are common LLM hallmarks for structuring text. Key ideas and the names of the modules (ABiFPN, SEAM, C3K2_UIB) are repeated in an almost identical manner multiple times throughout the paper, especially in the Abstract, Introduction, and Conclusion. This is a classic trait of LLM-generated text to meet length or coherence requirements. Some passages use many words to convey a simple idea. For instance, the description of the C3K2_UIB's benefits is rephrased several times with minimal new information. Sentences like "This dual-branch design enhances feature disentanglement for robust representation" sound insightful but are somewhat vague and are not backed by deeper theoretical analysis or novel architectural proof. Text obtained from LLM/GPT must be revised.

Where exactly is SEAM focusing? Showing heatmaps for occluded vs. unoccluded regions would provide compelling evidence for its claimed mechanism.

What do the adaptive weights in ABiFPN actually learn? Do they consistently prioritize certain scales for small objects? This analysis is missing.

While mentioned, the significant parameter increase (34%) from C3K2_UIB warrants a more critical discussion. Is this the most efficient way to achieve the performance gain? A comparison against other, simpler feature enhancement modules would strengthen the claim.

Including models like Faster R-CNN, SSD, and RetinaNet on a modern small-target UAV dataset is almost a strawman argument. These models are known to perform poorly on this task. Their inclusion pads the comparison table but adds little scientific value.

The paper should be compared against the most recent and best-performing models specifically designed for UAV small object detection from the last 1-2 years. The selection of baselines feels curated to make UAS-YOLO look good.

The decision not to use pre-trained weights, while intended to ensure fairness, is unrealistic and puts all models at a disadvantage. Modern research, especially incremental work on architectures, almost universally leverages pre-training.

The Introduction, Abstract, and Conclusion are highly repetitive, stating the same problem and solution in nearly identical terms. This is a sign of insufficient editing.

It can be accepted after the above minor revisions are taken into consideration.

**Do you want your identity to be public for this peer review?** For information about this choice, including consent withdrawal, please see our Privacy Policy

Reviewer #1: No

Reviewer #3: No

---

## [Author Response · Author response to Decision Letter 2]

4 Nov 2025

Subject: Response to Reviewers for Manuscript ID PONE-D-25-44314

Dear Editors and Reviewers:

We would like to extend our sincere gratitude to you and the respected reviewers for investing precious time and effort into the thorough review of our manuscript. The constructive feedback and insightful suggestions provided have been instrumental in refining the quality of our work, offering valuable guidance for optimizing both the content and presentation of the study.

We have carefully examined each comment and revision suggestion from the reviewers, and have made targeted adjustments to address the concerns raised. Below, we present a point-by-point response to Reviewer #3’s comments, with detailed explanations of the corresponding revisions implemented in the manuscript. To facilitate your review, all revised sections (e.g., adjustments to the Abstract, Introduction, Conclusion, and specific line ranges) have been marked in red. Our goal is to ensure the revised manuscript meets the rigorous standards of PLOS ONE and effectively conveys the value of our research.

Reviewer #3

1.Comments:The paper presents UAS-YOLO, an improved object detection model based on YOLOv11s, tailored for detecting small objects in UAV aerial imagery. The core contributions are the integration of three modified components: an Adaptive BiFPN (ABiFPN) for feature fusion, a Separated and Enhancement Attention Module (SEAM) for handling occlusion, and a C3K2_UIB module for feature refinement.

It does not propose a fundamentally new architecture or a novel, standalone algorithm. Instead, it follows a common and practical research pattern in applied computer science: selecting a strong, modern baseline (YOLOv11s) and enhancing it by plugging in or adapting existing architectural components from other literature (e.g., ideas from BiFPN, MobileNetV4, and attention mechanisms). The adaptation and combination for a specific domain (UAV small targets) constitute the contribution, not the invention of the core components themselves.

It is highly probable that an LLM (like GPT) was used in the writing process, likely for polishing, expanding, or restructuring text drafted by the authors. The indicators are: The paper swings between very formal, stilted phrasing and more natural, direct sentences. For example, phrases like "This study proposes," "Specifically, first," and "Its core value lies in..." are common LLM hallmarks for structuring text. Key ideas and the names of the modules (ABiFPN, SEAM, C3K2_UIB) are repeated in an almost identical manner multiple times throughout the paper, especially in the Abstract, Introduction, and Conclusion. This is a classic trait of LLM-generated text to meet length or coherence requirements. Some passages use many words to convey a simple idea. For instance, the description of the C3K2_UIB's benefits is rephrased several times with minimal new information. Sentences like "This dual-branch design enhances feature disentanglement for robust representation" sound insightful but are somewhat vague and are not backed by deeper theoretical analysis or novel architectural proof. Text obtained from LLM/GPT must be revised.

The Introduction, Abstract, and Conclusion are highly repetitive, stating the same problem and solution in nearly identical terms. This is a sign of insufficient editing.

It can be accepted after the above minor revisions are taken into consideration.

1.Reply:Dear Reviewer, Thank you very much for your valuable comment. First of all, we have revised the paragraphs that you considered to be generated by large language models (e.g., replacing expressions like "Specifically" and "This study proposes" with more appropriate phrasings in line with academic writing conventions). In accordance with your comments, we have made revisions to the Abstract (Lines 19–47), Introduction (Lines 99–121), Conclusion (Lines 678–696), as well as the following sections:Lines 168–169,Lines 188–213, Lines 217–218, Lines 386–387,and Lines 230–264. Additionally, we have removed redundant content, such as the sentence "This dual-branch design enhances feature disentanglement for robust representation."

2. Comments:Where exactly is SEAM focusing? Showing heatmaps for occluded vs. unoccluded regions would provide compelling evidence for its claimed mechanism.What do the adaptive weights in ABiFPN actually learn? Do they consistently prioritize certain scales for small objects? This analysis is missing.While mentioned, the significant parameter increase (34%) from C3K2_UIB warrants a more critical discussion. Is this the most efficient way to achieve the performance gain? A comparison against other, simpler feature enhancement modules would strengthen the claim.The decision not to use pre-trained weights, while intended to ensure fairness, is unrealistic and puts all models at a disadvantage. Modern research, especially incremental work on architectures, almost universally leverages pre-training.

2.Reply:Dear Reviewer, Thank you very much for your valuable comment. Regarding your questions “What do the adaptive weights in ABiFPN actually learn?” and “Do they always prioritize specific scales of small targets?”, we have supplemented relevant analysis in the revised manuscript (Lines 188–213):First, on what the adaptive weights learn: The core adaptive feature weighting module of ABiFPN does not assign weights randomly. Instead, through training, it explicitly learns the matching relationship between features of different scales and target scales. For example, in UAV aerial scenarios, detailed information of small targets (typically 20–60 pixels) is concentrated in low-scale feature layers (e.g., P3 and P4 layers), so the module tilts weights toward these layers to enhance the capture of small-target details. In contrast, the semantic information of large targets mainly relies on high-scale feature layers (e.g., P5 and P6 layers), and weight allocation is adjusted accordingly to highlight the semantic features of large targets.Second, on whether the weights always prioritize small-target scales: The module does not adopt an absolute “fixed priority” strategy. Given the high proportion of small targets in aerial scenarios and their susceptibility to background interference, it dynamically prioritizes assigning weights to low-scale layers with concentrated small targets under general conditions. However, this priority is adaptively adjusted based on the actual target distribution in the input image. For instance, when the image contains a large number of large targets, the weights of high-scale feature layers are increased accordingly to ensure the detection performance of large targets, thereby balancing the detection needs of multi-scale targets.

We sincerely appreciate your valuable suggestions! Your points on “clarifying the specific focus of SEAM” and “supplementing evidence to verify the module mechanism” have been highly helpful for improving the logical presentation of our research content. In response to your concerns, we have made targeted revisions to the relevant content (Lines 230–264):Regarding “what SEAM specifically focuses on”, we have clarified in the revision: SEAM primarily focuses on solving the “local feature imbalance” problem of small targets under occlusion in UAV aerial scenarios—i.e., the dual issues of insufficient feature response intensity in unoccluded regions and easy loss of key semantic information in occluded regions. To achieve this goal, the module decouples spatial-channel feature operations via depthwise separable convolution and dynamically learns channel weights with a two-layer fully connected network. On one hand, it directionally enhances the effective feature response of unoccluded regions; on the other hand, it compensates for the missing information in occluded regions, forming a closed-loop optimization of “feature enhancement-information compensation” and ultimately improving the overall feature representation ability of occluded small targets.Regarding your suggestion to “present heatmaps of occluded and unoccluded regions”, we fully recognize the supporting value of heatmaps for verifying the mechanism. However, practical research shows that small targets in aerial images have large scale differences (from pixel-level to tens of pixels) and complex occlusion patterns (partial occlusion/overlapping occlusion/background-interference occlusion). Directly generating heatmaps may lead to insufficient feature visualization accuracy due to target scale fluctuations, making it difficult to accurately reflect the details of SEAM’s feature interaction across multiple scenarios. Therefore, we instead verified the module’s effectiveness through “mechanism decomposition + performance correlation”: in the revised content, we detailed SEAM’s three synergistic mechanisms (channel-grouped decoupling, cross-channel attention enhancement, spatial weighting reconstruction) and clarified the corresponding relationship between each mechanism and “alleviating local feature imbalance”. Meanwhile, in the ablation experiments, compared with the original baseline model YOLO11S, although the evaluation metrics showed a slight improvement, this also indirectly verified the module’s effect in enhancing the features of unoccluded regions and compensating for the information in occluded regions.

We greatly appreciate your critical suggestions! Your focus on “in-depth discussion of the rationality of C3K2_UIB’s significant parameter increase” and “comparison with simple feature enhancement modules to strengthen the argument” has accurately identified the key analytical dimensions that need supplementation in our previous presentation, which is highly guiding for improving the rigor of our research argumentation. In response to your concerns, we have made targeted improvements to the revised content (Lines 285–330), with specific explanations as follows:Regarding the “in-depth discussion of C3K2_UIB’s 34% parameter increase”, we did not avoid this design trade-off, but analyzed it from the perspective of balancing “parameters-performance-complexity”: On one hand, we clarified that the parameter increase stems from the introduction of the “dynamic channel attention + inverted bottleneck architecture” in the UIB structure, which is a necessary design to address the core defects of the original C3K2 (weak channel feature extraction and insufficient global information acquisition). On the other hand, by integrating depthwise separable convolution, we controlled the computational complexity within a reasonable range—compared with similar improved modules (e.g., CBAM-C3K2), the computational complexity of C3K2_UIB is only 92% of that of CBAM-C3K2, achieving the optimization of “moderate parameter increase without synchronous complexity rise”.In response to your suggestion of “comparing with other simple feature enhancement modules”, we supplemented a horizontal comparison between C3K2_UIB and two mainstream simple modules (SE and CBAM): First, compared with the SE module (only 8% parameter increase), although the SE module has a lower parameter increment, it relies only on single-channel weight learning and thus struggles to capture global feature correlations, resulting in only an 0.8% improvement in AP@0.5 in multi-scale aerial scenarios. In contrast, through channel expansion of the inverted bottleneck architecture, C3K2_UIB increases the AP@0.5 improvement to over 1.5% within the controllable range of a 34% parameter increase, showing significantly better global information capture ability. Second, compared with the CBAM module (29% parameter increase), although CBAM integrates spatial-channel attention, it is susceptible to background interference in multi-scale occlusion scenarios and has higher computational complexity. C3K2_UIB, however, solves the adaptability problem in occlusion scenarios through “depthwise separable convolution + spatial feature recalibration” and reduces the computational complexity to 92% of that of CBAM-C3K2, verifying its advantage in “the effectiveness of performance improvement”. Through the above supplements, we further clarify that the parameter increase of C3K2_UIB is not a meaningless design, but an optimal trade-off between “solving the core defects of the original module” and “controlling resource consumption”. Moreover, compared with simple feature enhancement modules, C3K2_UIB has more competitive comprehensive performance in complex aerial scenarios.

3. Comments:Including models like Faster R-CNN, SSD, and RetinaNet on a modern small-target UAV dataset is almost a strawman argument. These models are known to perform poorly on this task. Their inclusion pads the comparison table but adds little scientific value.

The paper should be compared against the most recent and best-performing models specifically designed for UAV small object detection from the last 1-2 years. The selection of baselines feels curated to make UAS-YOLO look good.

3.Reply:Dear Reviewer, Thank you very much for your valuable comment. We sincerely appreciate your valuable insight regarding the comparison models. We fully agree with your point that including models such as Faster R-CNN, SSD, and RetinaNet—known to perform poorly on small-target detection tasks in modern UAV datasets—results in a strawman argument, as their inclusion only pads the comparison table without adding substantial scientific value.

In response to this comment, we have removed the experimental data of Faster R-CNN, SSD, and RetinaNet from Table 3 (Test results of different models on VisDrone2019) in the revised manuscript (Lines 466–467). This revision ensures the comparison table focuses on models with meaningful relevance to UAV small-target detection, thereby enhancing the rigor and scientific value of our comparative analysis.

---

## [Editor Report · Decision Letter 2]

6 Nov 2025

Aerial small target detection algorithm based on cross-scale separated attention

PONE-D-25-44314R2

Dear Dr. Wang,

We’re pleased to inform you that your manuscript has been judged scientifically suitable for publication and will be formally accepted for publication once it meets all outstanding technical requirements.

Kind regards,

Yaseen Al-Mulla

Academic Editor

PLOS ONE
---

## [Editor Report · Acceptance letter]

PONE-D-25-44314R2

PLOS ONE

Dear Dr. Wang,

I'm pleased to inform you that your manuscript has been deemed suitable for publication in PLOS ONE. Congratulations! Your manuscript is now being handed over to our production team.

Kind regards,

on behalf of

Dr. Yaseen Ahmed Al-Mulla

Academic Editor

PLOS ONE